# FedGain: Toward Negative-Gain-Free Client Collaboration in Federated Learning

**Yuqing Zhang** [1]  **Changli Zhou** [1]  **Binghuang Huang** [1]  **Hui Tian** [1]

## Abstract

Data heterogeneity is a fundamental challenge in Federated Learning (FL), where induced model drift often results in "negative gains" for global models on data-abundant clients, with performance falling below that of local training. To address this issue, we propose FedGain, a novel framework that optimizes collaborative client clustering to mitigate the negative gain. We are the first to develop a modified Scaling Law (SL) to quantify the reduction in data utility caused by heterogeneity and define Effective Federated Capacity to align clients with the highest potential collaboration gains. Extensive experiments demonstrate that our modified SL strictly adheres to the power-law learning discipline in non-IID scenarios. FedGain effectively suppresses negative gains to a negligible level across various FL algorithms and outperforms other Clustered FL methods.

## 1. Introduction

Federated learning (FL) is a distributed framework that enables multiple clients to collaboratively train a model via a central server without data leaving local devices. In cross-organization FL, participants are often heterogeneous in both data distribution and volume, such as hospitals with different patient types and volumes (Zhu et al., 2025). The core idea is that by aggregating multi-source knowledge (e.g., combining records from children's and maternity hospitals), the global model should theoretically surpass any individual local model. This expectation aligns with the widely observed Neural Scaling Laws (NSL) in deep learning, where model performance typically follows a power-law improvement with increasing data scale (Kaplan et al.,

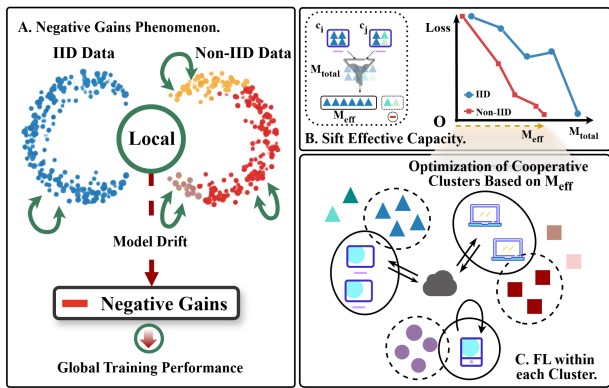

*Figure 1.* Negative Gains and Optimization of Client Collaboration.

2020).

However, unlike the idealized assumption of independent and identically distributed (IID) data, real-world federated clients inherently exhibit statistical heterogeneity, manifested as distribution shift (e.g., feature or label shift) and data quantity skew (e.g., sample size imbalance) (Ye et al., 2023). This heterogeneity often leads to model drift, where the models obtained by clients participating in FL may even underperform their locally trained models, referred to as negative gains. We identify this counter-intuitive phenomenon as a "Broken Neural Scaling Laws" in FL (Caballero et al., 2022): that is, simply increasing the total number of physical samples across the federation does not guarantee performance gains and may even result in performance stagnation or degradation.

Clustered Federated Learning (CFL) (Sattler et al., 2020; Long et al., 2023; Kim et al., 2024) serves as an effective approach to mitigating negative gains. By grouping clients with similar data distributions into clusters for independent federated training, CFL facilitates the sharing of relevant knowledge within clusters while minimizing negative interference from distribution shifts. By optimizing the collaborative relationships among clients, CFL enhances training efficiency and effectively alleviates model drift and avoids negative gains without introducing significant overhead.

In this context, the core challenge in CFL for mitigating negative gains lies in identifying optimal collaborative relationships. However, conventional methods exhibit signif-

[1]Department of Computer Science and Technology, Huaqiao University, Xiamen, China. Correspondence to: Changli Zhou <zcl@hqu.edu.cn>.

*Proceedings of the $43^{rd}$ International Conference on Machine Learning*, Seoul, South Korea. PMLR 306, 2026. Copyright 2026 by the author(s).

icant limitations. Most existing works rely predominantly on distribution distances, such as model parameters or loss discrepancies, while neglecting the impact of data quantity imbalances on update directions and convergence stability. For instance, methods like FeSEM (Long et al., 2023) and IFCA (Ghosh et al., 2020) cluster clients solely by model similarity or loss without accounting for data volume. This can marginalize small-scale clients during aggregation, failing to sufficiently alleviate negative gains in highly heterogeneous and imbalanced federated scenarios.

Consequently, we attribute model performance degradation to two pivotal factors: distribution shift and quantity skew. Generally, greater distributional divergence reduces collaborative relevance, while data imbalance further exacerbates update deviation, leading to model tilting.

While Scaling Laws (SL) accurately characterize model behavior in overparameterized regimes, they fail in FL by neglecting data heterogeneity (Kaplan et al., 2020). Currently, theoretical tools lack the precision to quantify how distributional divergence erodes the generalization gains of data scale. For distribution shift estimation, methods like FedEM (Marfoq et al., 2021) often trap in local optima by simultaneously optimizing parameters and structures. Meanwhile, approaches using pairwise discriminators incur excessive overhead and locality issues (Bao et al., 2023). Both paradigms result in suboptimal clustering performance.

To address these challenges, we propose FedGain, a framework that optimizes client collaboration to mitigate negative gains. First, we derive a FL error bound of tightly coupling distribution shift with data scale. Through federated contrastive learning, we estimate the global distribution shift distance without violating FL privacy constraints. We posit that the generalization performance of FL is not determined by the total number of physical samples, but rather by the Effective Federated Capacity (EFC). Thus, we are the first to derive a heterogeneity-aware generalization bound under modified SL, enabling the identification of beneficial data for theoretical clustering guidance. Second, we design a structure optimization method that is decoupled from the training of the federated classification task. Unlike traditional CFL, which requires structure optimization to run throughout the entire training process, our method can rapidly solve for the optimal client collaboration topology. Finally, FedGain decouples structure optimization from model training, allowing seamless integration with any Global Federated Learning (GFL) or Personalized Federated Learning (PFL) algorithm to enhance performance without additional training overhead.

Our contributions are summarized as follows:

- We introduce the concept of effective federated capacity, establish a federated scaling law for non-IID

settings, and derive a client error bound that tightly couples distribution shift with quantity skew.

- Guided by this theory, we propose FedGain to achieve optimal client collaboration structure optimization that is decoupled from the federated task training.

- Extensive experiments across diverse datasets, models, and heterogeneity settings validate our approach. FedGain not only restores the validity of scaling laws in heterogeneous FL but also consistently enhances SOTA GFL and PFL methods by avoiding negative gains. Furthermore, FedGain outperforms existing CFL baselines in both predictive accuracy and clustering quality.

## 2. Related Work

Model drift, rooted in the inconsistency between local distributions and global objectives, remains a fundamental challenge in heterogeneous FL. Early GFL methods, such as FedProx (Li et al., 2020) and SCAFFOLD (Karimireddy et al., 2020), employ training constraints or control variates to regularize local updates, while FedNova (Wang et al., 2020) normalizes step sizes to correct aggregation bias. However, these approaches force a single model to fit across non-IID data without addressing root-cause heterogeneity. Alternatively, PFL tailors parameters to local conditions via meta-learning (Per-FedAvg (Fallah et al., 2020)), multi-task learning (Ditto (Li et al., 2021)), or knowledge distillation (FedAS (Yang et al., 2024)). While these "circumvent" drift, they lack explicit collaborative structure selection, often forcing knowledge sharing between mismatched clients and triggering implicit negative transfer (Zhang et al., 2021; Bao et al., 2023).

Similar to our work, CFL actively mitigates drift by forming homogeneous collaborative clusters. Existing CFL methods primarily start from the perspective of data distribution divergence, attempting to construct more reasonable training clusters through client clustering or multi-task learning. For instance, IFCA (Ghosh et al., 2020) and FeSEM (Long et al., 2023) perform alternating clustering and model updates based on loss values or distances in parameter space or FedEM (Marfoq et al., 2021) learns personalized mixture weights. However, they all overlook the impact of data quantity variance on model updates. In contrast, we decouple and then recouple distribution shift and quantity skew into a non-linear Effective Capacity metric, enabling precise correction of heterogeneous drift by accounting for both factors simultaneously.

Data heterogeneity-induced model drift is the primary driver of performance degradation in FL. While traditional analyses based on VC-dimension or stability (Liu et al., 2015) often yield loose bounds, Neural Scaling Laws (Kaplan et al., 2020) offer a more precise power-law framework for over-

parameterized models. However, existing NSL research is largely restricted to centralized IID settings (Caballero et al., 2022), making it ill-suited for the data heterogeneity inherent in FL. This paper revises the scaling law for the federated setting by replacing the total physical data volume with the effective data volume attenuated by heterogeneity. This provides a quantitative basis for generalization gains under coupled distribution shift and quantity skew. Guided by this theory, we propose FedGain to optimal client cooperation structures, significantly improving the performance of existing SOTA methods.

## 3. Theoretical Analysis

### 3.1. Setup

In a given FL system, assume that there are $N$ clients ($i \in 1, 2, \cdots, N$) participating in the training process via a central server. Each client possesses its own local dataset $\mathcal{D}_i$ with a distinct data distribution $P_i$, and holds $m_i$ samples drawn from its true local data distribution $P_i$. Consequently, the heterogeneous data distribution for each client $i$ is represented as $P_i = \{x_k^{(i)}, y_k^{(i)}\}_{k=1}^{m_i}$, where $x_k^{(i)}$ denotes the features of the $k$-th sample on client $i$ and $y_k^{(i)}$ is the corresponding label. Let the total number of samples across all participating clients be $M$. The proportion of samples held by each client is given by the vector $p = [p_1, p_2, \cdots, p_n] = [\frac{m_1}{M}, \frac{m_2}{M}, \cdots, \frac{m_n}{M}]$, satisfying $\sum_{i=1}^{N} p_i = 1$. In practice, the primary objective for each client in federated training is to minimize its local training loss. Given a machine learning model $h$ and a loss function $l$ specified by the FL system, the expected loss of client $i$'s true data distribution on the model $h$ is defined as $r_l(h) = E_{(x,y) \sim P_l}[l(h(x), y)]$. The empirical loss, which approximates the expected loss using the finite local dataset, is given by $\hat{r}_i(h) = \frac{1}{m_i} \sum_{k=1}^{m_i} l(h(x_k^{(i)}), y_k^{(i)})$. Each client $i$ aims to find a model h within a given hypothesis space $\mathcal{H}$ that minimizes its expected risk. This optimal model for client $i$ is denoted as $h_i^* = \underset{h \in \mathcal{H}}{\operatorname{argmin}} \hat{r}_i(h)$.

During local training, a client can only utilize its limited local samples to optimize the model, i.e., the local data is used to optimize $\hat{h}_i = \underset{h \in \mathcal{H}}{\operatorname{argmin}} \hat{r}_i(h)$. In global training, the information from all clients is aggregated according to their respective weights $\alpha_i$, and a single global model is trained to minimize the weighted average of all clients' local losses (McMahan et al., 2017), i.e., the client optimizes $\hat{h}_{i,\alpha_i} = \underset{h \in \mathcal{H}}{\operatorname{argmin}} \alpha_i \cdot \hat{r}_i(h)$.

Clustered training typically serves as a compromise approach between fully local training without collaboration and fully global training with overall collaboration. Clusters can be regarded as a coalition of cooperating clients, forming a collaborative cluster structure $C = [C_1, C_2, \cdots, C_k]$, where the weights of clients within cluster $C_k$ are denoted as $\alpha_k = [\alpha_{k1}, \alpha_{k2}, \cdots, \alpha_{kn}]$. The optimization objective for clients within each cluster follows the GFL formulation: $\hat{h}_{i,\alpha_k} = \underset{h \in \mathcal{H}}{\operatorname{argmin}} \alpha_{k,i} \cdot \hat{r}_k(h)$, where client $i$ belongs to collaborative cluster $j$. Specifically, CFL partitions all participating clients into several clusters based on their data distributions. Federated training is then conducted within each cluster by training a cluster-specific model to minimize the weighted average of the local losses of all clients within that cluster.

Consequently, an ideal collaborative structure should enable intra-cluster distributions to converge toward IID, thereby eliminating the adverse effects of heterogeneity. Unlike existing methods that rely on alternating "train-and-cluster" passive iterations, we aspire to perform privacy-friendly distribution inference to quantify inter-client relationships. This would allow for the proactive construction of an optimal collaborative structure prior to training, avoiding high iterative costs and the risk of local optima.

### 3.2. Theoretical Error Bound

Data heterogeneity that induces negative gains can be categorized based on its statistical properties (Ye et al., 2023) into two primary factors: distribution shift and quantity bias. Accordingly, we will derive a comprehensive error bound for clients by incorporating these two influencing factors.

**Assumption 1 (Effective Federated Capacity).** We posit that the generalization performance of a federated model is determined not by the total physical sample size $M_{total}$, but by the effective sample size $M_{eff}$ after being attenuated by distribution heterogeneity. For a target client $i$, the contribution of samples $m_j$ from another client $j$ should be weighted by their distribution distance $D(P_i, P_j)$. That is:

$$M_{eff}^{(i)} = m_i + \sum_{j \neq i} \beta(D(P_i, P_j)) \cdot m_j \qquad (1)$$

Where $\beta(\cdot)$ is a monotonically decreasing kernel function with respect to the distribution distance (Hofmann et al., 2008; Wang et al., 2017). This assumption forms the basis for the subsequent **Heterogeneity-aware Federated Scaling Law** (HFSL). Within CFL framework, we partition clients into collaborative clusters $C = [C_1, C_2, \cdots, C_k]$, aiming to maximize the effective federated capacity within each cluster by optimizing the collaborative structure, thereby approaching the local optimum.

**Definition 3.1 (Distribution Shift).** To adhere to the privacy-friendly principle, we employ contrastive learning (Oord et al., 2018) to train a shared feature extractor $\phi : \chi \rightarrow R^d$ for indirectly measuring distribution shift. The optimization objective is to maximize the mutual informa-

tion between positive sample pairs $(x, x^+)$ within a batch $\mathcal{B}$:

$$
\begin{aligned}
\min_\phi \mathcal{L}_c = -\mathbb{E}_{(x,x^+)\sim P_i^{\text{aug}}} \Big[ & \phi(x)^T \phi(x^+)/\tau \\
& - \log \sum_{x^- \in \mathcal{B}} \exp(\phi(x)^T \phi(x^-)/\tau) \Big]
\end{aligned}
\tag{2}
$$

Where $P_i^{\text{aug}}$ denotes the augmented data distribution derived from $P_i$, and $\tau$ is a temperature hyperparameter. Each client $i$ maps its local data to a set of feature vectors $\phi(x_k^{(i)})_{k=1}^{m_i}$ via the optimized feature extractor $\phi$. Based on the $\mathcal{H}\Delta\mathcal{H}-divergence$ (Ben-David et al., 2010), we define the distribution shift $D(P_i, P_j)$ between clients $i$ and $j$ as the normalized distance between their feature centroids:

$$
D(P_i, P_j) = 1 - \frac{\mathbb{E}_{P_i}[\phi(x)]^T \mathbb{E}_{P_j}[\phi(x)]}{\|\mathbb{E}_{P_i}[\phi(x)]\| \cdot \|\mathbb{E}_{P_j}[\phi(x)]\|}
\tag{3}
$$

In this normalized feature space, this distance satisfies the *L-Lipschitz* continuity condition, leading to an upper bound on the risk induced by distribution shift:

$$
|r_i(h) - \hat{r}_i(h)| \le L \cdot D(P_i, P_j)
\tag{4}
$$

**Definition 3.2 (Generalization Error Dependent on Data Volume).** Traditional NSL (Kaplan et al., 2020) posits that model test error follows a power-law relationship with training data volume: $L(M) \propto M^{-\nu}$. However, this law becomes invalid under heterogeneous data in FL. We propose the **Heterogeneity-aware Federated Scaling Law** (HFSL): in a non-IID federated cluster $\mathcal{C}_k$, the data-dependent generalization error bound $\Psi$ is no longer determined by the total physical sample count $M_k = \sum_{j \in \mathcal{C}_k} m_j$, but by the **Effective Federated Capacity**. For any client $i$ in cluster $\mathcal{C}_k$, given a confidence parameter $\delta \in (0, 1)$, there exists a scaling exponent $\nu > 0$ and a constant $C$ such that the generalization error $\Psi$ satisfies the following inequality with probability at least $1 - \delta$:

$$
|r_{C_k}(h) - \hat{r}_{C_k}(h)| \le \Psi(\alpha, m, D, \delta)
\tag{5}
$$

Specifically, the HFSL-based generalization error bound is formalized as

$$
\begin{aligned}
\Psi_i(\alpha, m, D, \delta) = C \cdot \Big( & m_i + \sum_{j \in C_k, j \neq i} \alpha_j \cdot m_j \\
& \cdot e^{-\gamma D(P_i, P_j)} \Big)^{-\nu}
\end{aligned}
\tag{6}
$$

Where $\nu$ is the scaling exponent (Hestness et al., 2017).By introducing the decaying kernel $e^{-\gamma D(\cdot)}$, this law captures the phenomenon that the marginal contribution of low-quality data (with high distribution shift) to generalization performance diminishes, or even vanishes.

**Theorem 3.3 (Error Bound).** For a cluster $\mathcal{C}_k$, let $h_i^*$ be the theoretical optimal model for client i. Combining **Definitions 3.1 and 3.2**, for any $\delta \in (0, 1/2)$, the expected risk of client $i$ using the cluster model $h_{\mathcal{C}_k}$ satisfies the following upper bound with probability at least $1 - 2\delta$:

$$
\begin{aligned}
r_i(h_{\mathcal{C}_k}) \le r_i(h_i^*) + 2 \sum_{j \in C_k, j \neq i} \alpha_j D(P_i, P_j) \\
+ 2\Psi_i(\alpha, m, D, \delta)
\end{aligned}
\tag{7}
$$

Here, $r_i(h_i^*)$ is the irreducible minimum local expected risk for client $i$ given the hypothesis space $\mathcal{H}$. This theorem indicates that the optimal collaborative structure must strike a balance between "finding partners with similar distributions (reducing $D$)" and "expanding the effective data volume (reducing $\Psi$)". It reveals the fundamental trade-off in federated collaboration: the optimal structure should not blindly pursue cluster size (physical sample count), but rather aim to maximize the effective capacity. When $D(P_i, P_j)$ is excessively large, the gain from the HFSL term becomes negligible, while the bias term increases rapidly, thereby suppressing ineffective collaboration. See Appendix A.2 for the positive gain condition corollary.

# 4. Recommended Method

## 4.1. Optimization Objective

We begin by neglecting $r_i(h_i^*)$ term; then transform the theoretical expected error into a tractable empirical estimate.

**Distribution Shift Estimation via Contrastive Learning.** The theoretical $D(P_i, P_j)$ is difficult to compute directly. Previous CFL works (Sattler et al., 2020; Long et al., 2023) used cosine distance or distances between models to indirectly estimate distribution distance. However, due to high client heterogeneity and the non-convexity of neural networks, these measurements often suffer from significant variance. Leveraging contrastive learning, which maps data into a unified feature space, the distribution of client $i$ can be approximated by the mean of its feature vectors $\mu_i = \mathbb{E}_{x \sim P_i}[\phi(x)] = \frac{1}{m_i}\sum_{k=1}^{m_i} \phi(x_k^{(i)})$. The distribution shift can then be empirically estimated as the cosine distance between these class feature centroids:

$$
\widehat{D}(P_i, P_j) = 1 - \frac{\mu_i^T \mu_j}{\|\mu_i\| \cdot \|\mu_j\|}
\tag{8}
$$

Since only the class feature centroids $\mu_i$ are exchanged, this process strictly adheres to the privacy constraints in FL.

**Estimating Generalization Error via HFSL.** To overcome the limitations of traditional approaches, we directly adopt the HFSL generalization bound as the optimization term. We consolidate the constant terms and consolidate the constant terms from Theorem 3.3.

To formulate the global optimization objective, we aim to solve for an assignment matrix $A \in \{0,1\}^{N \times K}$, where $\alpha_{ij} = 1$ indicates that client $i$ joins cluster $j$. Unlike prior works that treat "distance" and "quantity" as two separate additive terms, the HFSL framework couples them into a unified nonlinear objective. The global loss function is constructed as follows:

$$\mathcal{L}(A, \boldsymbol{m}, \widehat{D}) = \sum_{k=1}^{K} \sum_{i \in \mathcal{C}_k} \frac{C}{\left( m_i + \sum_{j \in \mathcal{C}_k, j \neq i} m_j \cdot e^{-\gamma \widehat{D}_{ij}} \right)^{\nu}}$$

(9)

In this objective, minimizing the loss is equivalent to maximizing the effective federated capacity for each client. If a client j with a vastly different distribution is forcibly added to a cluster, although the physical sample size increases, the effective capacity remains almost unchanged due to $e^{-\gamma D_{ij}} \to 0$, which cannot offset the introduced bias risk. Notably, our clustering objective serves as a principled relaxation of a more rigorous global optimization goal.

### 4.2. Optimizer and Algorithm

Since the assignment matrix $A$ is discrete, direct optimization via gradient descent is infeasible (Jang et al., 2017). To address this, we introduce soft-assignment collaborative optimization by relaxing $A$ into a soft assignment matrix $S \in \mathbb{R}^{N \times K}$, where $s_{ik} \in [0,1]$ denotes the probability that client $i$ belongs to cluster $k$. Consequently, the Expected Effective Capacity obtained by client $i$ in cluster $k$ becomes:

$$M_{eff}^{(i,k)}(S) = m_i + \sum_{j \neq i} s_{ik} \cdot m_j \cdot e^{-\gamma \widehat{D}_{ij}}$$

(10)

So the differentiable global optimization objective $\mathcal{L}_s(S)$ after soft assignment is formulated as:

$$\mathcal{L}_s(S) = \sum_{k=1}^{K} \sum_{i=1}^{N} s_{ik} \Bigg( \frac{C}{\left( m_i + \sum_{j \neq i} s_{ik} \cdot m_j \cdot e^{-\gamma D_{ij}} \right)^{\nu}}$$
$$+ \sum_{j \neq i} s_{jk} \widehat{D}_{ij} \Bigg)$$

(11)

To ensure S remains a valid probability distribution matrix, it must satisfy $s_{ik} \leq 0$ and $\sum_{k=1}^{K} s_{ik} = 1$, forming a probability simplex. During optimization, we employ Projected Gradient Descent (PGD) on the simplex. Specifically, after each gradient update step $S' \leftarrow S - \eta \nabla_S \mathcal{L}_s$, we project each row of $S'$ onto the simplex:

$$S_{row(i)} \leftarrow Proj_{\Delta}(S'_{row(i)}) = \arg \min_{y \in \Delta} \| y - S'_{row(i)} \|_2^2$$

(12)

This projection ensures the constraints are satisfied and helps avoid local optima in the discrete space. Consequently, our

---

**Algorithm 1** FedGain

1: **Input:** Contrastive learning rounds $T_c$, collaboration rounds $T_s$, $N$ clients, dataset $\mathcal{D}_i$, sample size $m_i$.
2: Initialize feature extractor $\phi$, matrix $\mathbf{A} \in \mathbb{R}^{N \times K}$.
3: **for** $t = 0, \dots, T_c - 1$ **do**
4:      **for** each client $i$ in parallel **do**
5:          Send augmented class centroids to server
6:      **end for**
7:      Server computes $\mathcal{L}_c$ and $\phi \leftarrow \phi - \eta \nabla \phi_{\mathcal{L}_c}$
8: **end for**
9: Server computes feature centers $\mu_i = \mathbb{E}_{x \sim P_i}[\phi(x)]$
10: Update distribution shift distance matrix $\widehat{D}_{ij}$ by Eq. (8)
11: Quantity skew estimate $\widehat{\Psi}$ by Eq. (6)
12: **for** $t = 0, \dots, T_s - 1$ **do**
13:      Update error bound $r_i(h_{\mathcal{C}_k})$ by Eq. (7)
14:      Server projects $\mathbf{A}$ onto probability simplex $\mathbf{S} \leftarrow Proj_{\Delta(\mathbf{A})}$ and $\mathbf{S} \leftarrow \mathbf{S} - \eta \nabla S_{\mathcal{L}(\mathbf{S})}$
15:      $\mathcal{L} = \sum_{k=1}^{K} \sum_{i=1}^{N} s_{ik} \left( \frac{C}{(M_{eff}^{(i,k)}(\mathbf{S}))^{\nu}} + \sum_{j \neq i} s_{jk} \widehat{D}_{ij} \right)$
16: **end for**

---

approach can stably find a collaborative client structure close to the global optimum across various non-IID scenarios.

Note that the gradient term $\nabla_{s_{ij}} \mathcal{L}$ explicitly contains the factor $e^{-\gamma D_{ij}}$, which implies that the algorithm assigns higher weights to clients that are "data-rich and distributionally similar," while automatically suppressing those that are "data-rich but distributionally divergent." Mathematically, this ensures the system stably converges to the globally optimal collaborative structure, effectively mitigating the model drift problem in non-IID scenarios. Furthermore, our method leverages pre-estimated distribution characteristics to directly search for the optimal structure via PGD over the probability simplex, without requiring model training updates. Algorithm 1 presents the FedGain framework.

## 5. Experiments

### 5.1. Setup

**Datasets.** We employ the CIFAR10, FMNIST, and CI-FAR100 datasets (Krizhevsky et al., 2009; Xiao et al., 2017), simulating different distribution shift scenarios by applying the following configurations to enhance data heterogeneity across clients.

- **Feature Shift** (Ghosh et al., 2020). CIFAR10 images are rotated by 0°, 120°, and 240°, respectively. Clients are grouped by indices (0-4, 5-9, 10-14, and 15-19) and assigned distributions as follows: 100% 0° rotation, a 1:1 mixture of 0° and 120° rotations, a 1:1 mixture of 120° and 240° rotations, and a 1:2:2 mixture of all three rotations. In this setting, the feature distribution

*Table 1.* Performance enhancement and negative gain reduction of GFL and PFL methods across diverse heterogeneous scenarios. Means and standard deviations reported from 5 different random seeds.

| Method | CIFAR10 (Feature Shift) | | | FMNIST (Label Shift) | | | CIFAR100 (Quality Shift) | | |
|---|---|---|---|---|---|---|---|---|---|
| | ACC↑ | NGP↓ | RSD↓ | ACC↑ | NGP↓ | RSD↓ | ACC↑ | NGP↓ | RSD↓ |
| Local Train | 35.63 (2.48) | - | - | 85.63 (0.07) | - | - | 29.18 (0.18) | - | - |
| FedAvg | 44.74 (0.27) | 10.00 (0.00) | 4.80 (0.60) | 46.62 (0.06) | 55.00 (5.00) | 40.85 (0.05) | 26.27 (0.05) | 50.00 (0.00) | 12.56 (1.90) |
| + FedGain | **50.44** (0.36) | **0.00** (0.00) | **3.20** (0.64) | **90.16** (0.59) | **0.00** (0.00) | **5.29** (1.13) | **37.95** (0.82) | **0.00** (0.00) | **7.32** (2.69) |
| FedProx | 45.34 (0.54) | 10.00 (0.00) | 5.08 (0.08) | 46.54 (0.05) | 60.00 (5.00) | 40.69 (0.09) | 26.57 (0.45) | 50.00 (0.00) | 12.79 (2.04) |
| + FedGain | **51.63** (1.21) | **0.00** (0.00) | **3.03** (0.34) | **90.20** (0.61) | **0.00** (0.00) | **5.34** (1.10) | **35.64** (1.35) | **0.00** (0.00) | **8.30** (2.53) |
| FedNova | 47.35 (0.28) | 7.50 (2.50) | 4.03 (1.26) | 75.15 (0.72) | 60.00 (5.00) | 12.64 (0.71) | 26.09 (0.08) | 50.00 (0.00) | 10.82 (1.31) |
| + FedGain | **50.90** (0.70) | **0.00** (0.00) | **2.90** (0.40) | **90.12** (0.58) | **10.00** (0.00) | **5.39** (1.16) | **36.22** (2.31) | **0.00** (0.00) | **7.68** (2.12) |
| FedCSD | 40.86 (0.34) | 15.00 (5.00) | 4.82 (0.40) | 46.40 (0.07) | 65.00 (5.00) | 40.67 (0.28) | 26.55 (2.21) | 50.00 (0.00) | 9.83 (1.78) |
| + FedGain | **44.30** (1.23) | **0.00** (0.00) | **3.95** (0.26) | **90.36** (0.57) | **17.50** (7.50) | **5.49** (0.94) | **33.12** (3.72) | **5.00** (0.00) | **7.48** (2.59) |
| Finetune | 45.37 (0.95) | 7.50 (2.50) | 5.09 (0.32) | 46.59 (0.04) | 52.50 (2.50) | 40.82 (0.02) | 27.47 (0.06) | 50.00 (0.00) | 12.81 (1.33) |
| + FedGain | **49.14** (0.91) | **0.00** (0.00) | **2.94** (0.48) | **90.27** (0.47) | **10.00** (5.00) | **5.25** (1.25) | **36.76** (1.67) | **12.50** (2.50) | **9.82** (0.65) |
| Per-FedAvg | 43.61 (0.92) | 5.00 (5.00) | 10.39 (0.58) | 46.51 (0.05) | 57.50 (7.50) | 37.17 (1.73) | 25.89 (0.20) | 55.00 (5.00) | 14.56 (0.36) |
| + FedGain | **46.44** (1.41) | **0.00** (0.00) | **5.75** (0.33) | **90.35** (0.68) | **12.50** (2.50) | **19.02** (4.24) | **35.57** (0.03) | **12.50** (2.50) | **9.11** (2.52) |
| pFedMe | 38.00 (0.86) | 37.50 (2.50) | 9.32 (3.32) | 46.53 (0.06) | 57.50 (7.50) | 40.59 (0.07) | 25.81 (0.00) | 50.00 (0.00) | 13.20 (0.54) |
| + FedGain | **47.48** (1.45) | **0.00** (0.00) | **5.19** (2.07) | **90.02** (0.61) | **5.00** (0.00) | **5.44** (1.23) | **34.36** (1.50) | **5.00** (0.00) | **4.23** (0.25) |
| Ditto | 45.36 (0.61) | 10.00 (0.00) | 5.42 (0.13) | 46.70 (0.09) | 60.00 (5.00) | 40.89 (0.17) | 26.13 (0.35) | 50.00 (0.00) | 12.86 (1.43) |
| + FedGain | **49.25** (0.79) | **0.00** (0.00) | **2.92** (0.58) | **90.17** (0.62) | **0.00** (0.00) | **5.32** (1.16) | **35.42** (0.90) | **0.00** (0.00) | **8.32** (1.41) |
| FedAS | 45.66 (0.60) | 7.50 (2.50) | 4.03 (0.23) | 75.36 (0.33) | 70.00 (5.00) | 12.07 (0.48) | 27.53 (0.10) | 50.00 (0.00) | 15.03 (0.10) |
| + FedGain | **50.05** (0.44) | **0.00** (0.00) | **3.28** (0.10) | **89.05** (0.91) | **5.00** (0.00) | **5.95** (1.29) | **35.25** (1.50) | **0.00** (0.00) | **11.23** (1.00) |

distances between any pair of client groups are all different. Details are illustrated in Fig. 2-A.

- **Label Shift** (Ma et al., 2022)). The FMNIST dataset is partitioned across clients with varying data volumes and label distributions. Clients 0-4 and 5-9 have fewer samples with partially overlapping label sets. Clients 10-14 and 15-19 have larger sample sizes, also with partially overlapping labels. The label sets between clients 0-9 and clients 10-19 are non-overlapping, as shown in Fig. 2-B.

- **Quality Shift** (Ye et al., 2023). For the CIFAR100 dataset (using its 20 coarse-grained labels), we first apply label corruption. Then, to simulate quality shift with concept alteration. We add Gaussian noise ($\sigma$ = 0.1 to 0.4) to clients 0-4, 5-9, 10-14, and 15-19, respectively. Furthermore, different label mapping functions are applied: identity mapping, partial corruption mapping, full corruption mapping, and complex corruption mapping, as detailed in Fig. 2-C.

**Model.** First, a pre-trained ResNet18 (He et al., 2016) model is adopted to extract features from client data. As the parameters of this model remain independent of the federated task training process, no additional overhead is introduced to the subsequent federated training. Subsequently, for the three distribution scenarios constructed with CIFAR10, FMNIST,

and CIFAR100, we train a CNN model, an MLP model, and a ResNet18 model, respectively.

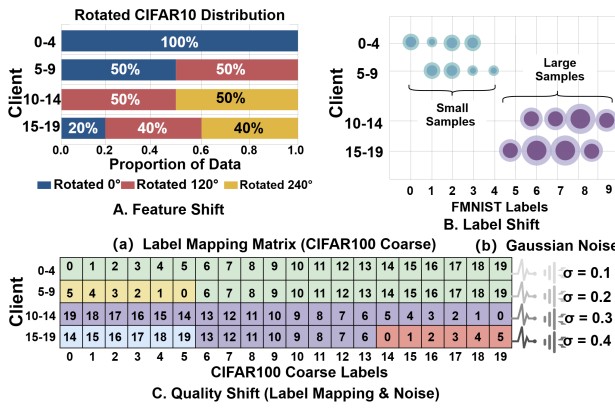

*Figure 2.* Data shift scenarios.

**Baselines.** We apply our FedGain method to optimize GFL methods, including FedAvg (McMahan et al., 2017), FedProx (Li et al., 2020), FedNova (Wang et al., 2020), and FedCSD (Yan et al., 2025). We also apply it to optimize PFL methods, including Finetune (where each client fine-tunes the FedAvg model locally), Per-FedAvg (Fallah et al., 2020), pFedMe (T Dinh et al., 2020), Ditto (Li et al., 2021), and FedAS (Yang et al., 2024). Furthermore, we compare it with CFL methods to evaluate the improvement our optimization objective brings to the clustering results.

**Performance Evaluation.** In addition to evaluating accu-

*Table 2.* Performance comparison with clustered federated learning baselines.

| Method | CIFAR10 (Feature Shift) | | | FMNIST (Label Shift) | | | CIFAR100 (Quality Shift) | | |
|---|---|---|---|---|---|---|---|---|---|
| | ACC↑ | NGP↓ | RSD↓ | ACC↑ | NGP↓ | RSD↓ | ACC↑ | NGP↓ | RSD↓ |
| IFCA | 48.77 (1.12) | 0.00 (0.00) | 3.42 (0.45) | 89.94 (0.62) | 10.00 (5.00) | 6.03 (1.12) | 28.82 (0.85) | 40.00 (5.00) | 12.34 (1.55) |
| FedCluster | 42.90 (0.95) | 20.00 (5.00) | 6.03 (0.88) | 90.11 (0.48) | 10.00 (2.50) | 6.51 (0.94) | 28.23 (1.02) | 35.00 (7.50) | 9.86 (1.20) |
| FeSEM | 42.14 (1.34) | 20.00 (7.50) | 4.90 (1.05) | 61.98 (2.15) | 55.00 (10.0) | 37.98 (2.41) | 29.78 (1.18) | 30.00 (5.00) | 10.48 (1.82) |
| CFL-GP | 40.81 (1.56) | 25.00 (10.0) | 6.72 (1.21) | 59.17 (3.04) | 85.00 (5.00) | 31.90 (4.12) | 28.62 (0.94) | 50.00 (0.00) | 11.95 (2.05) |
| **FedGain** | **50.44** (0.36) | **0.00** (0.00) | **3.20** (0.64) | **90.16** (0.59) | **0.00** (0.00) | **5.29** (1.13) | **37.95** (0.82) | **0.00** (0.00) | **7.32** (2.69) |

racy (ACC), we leverage the Incentive Participation Rate (IPR) (Cho et al., 2023) to derive the Negative Gain Proportion (NGP), which quantifies the fraction of clients suffering from performance degradation. Furthermore, we employ the Reward Standard Deviation (RSD) (Lyu et al., 2020) to evaluate the fairness of accuracy improvements across all clients relative to their respective local training. All metrics are defined using the client's local model $h_i^{local}$ and the FL model $h_i^{FL}$.

$$NGP = 1 - IPR = \frac{1}{N} \sum_{i=1}^{N} \mathbb{I}\left\{ acc(\hat{h}_i^{local}) - acc(\hat{h}_i^{FL}) > 0 \right\} \tag{13}$$

$$RSD = SD\left( \left\{ acc(\hat{h}_i^{FL}) - acc(\hat{h}_i^{local}) \right\}_{i=1}^{N} \right) \tag{14}$$

Where SD denotes standard deviation. In an ideal FL system free from model drift, all clients would obtain the same accuracy gain. This corresponds to a smaller NGP and a smaller RSD.

### 5.2. Performance Results

Experiments using NGP metrics signify varying degrees of negative gains in all GFL and PFL methods. However, integrating FedGain effectively alleviates these issues across SOTA baselines in three distinct distribution shift scenarios.

Table 1 demonstrates that our framework consistently mitigates negative gain across diverse datasets and models, significantly enhancing accuracy and fairness (RSD).

**Feature Shift.** Image rotation induces a relatively mild distribution shift, often serving as a form of data augmentation. Consequently, the NGP metric reveals that negative gains are less prevalent in this scenario, affecting only 0–20% of clients. FedGain further suppresses these remaining negative gains and boosts accuracy across all baselines, pushing the NGP toward 0% and concurrently enhancing system fairness.

**Label Shift.** NGP metrics reveal severe negative gains across all baselines, where both GFL and PFL models significantly underperform relative to local training. This degradation stems from GFL's inability to harmonize divergent distributions and the detrimental impact of the global model to personalized models in PFL. After optimization with the

FedGain, the negative gains are significantly alleviated. All methods achieve substantial accuracy improvements, approaching or surpassing local training performance, while fairness is also greatly enhanced.

**Quality Shift.** The training data with varying quality is simulated by introducing label misassignments and noise interference at the clients. As indicated by the NGP, all baseline methods suffer from severe negative gains: the updated models in both GFL and PFL perform poorly compared to locally trained models. After optimization with the FedGain, all methods achieve improved accuracy, surpassing local training performance, while fairness is also effectively enhanced. We observe that, for each client, whether in GFL or PFL, enabling effective collaboration among clients can mitigate negative gains present during training.

### 5.3. Comparison Results with Clustered FL

In CFL, client clustering is typically achieved through methods based on loss, gradient, or model parameter distance. For comparison, we employ baseline algorithms corresponding to these approaches: IFCA (Ghosh et al., 2020), FedCluster (Sattler et al., 2020), and FeSEM (Long et al., 2023), alongside CFL-GP (Kim et al., 2024) of a gradient-based soft clustering FL method distinct from traditional hard clustering. The FedGain is combined with FedAvg (McMahan et al., 2017) for cluster model training.

Table 2 presents the results. The soft clustering FL method CFL-GP performs less satisfactorily compared to the three hard clustering methods, indicating its inapplicability across the three classic non-IID scenarios. In all scenarios, the FedGain achieves the highest accuracy and fairness, while being the least affected by model drift. The primary reason is that FedGain considers quantity disparity in addition to distribution shift. In contrast, other baselines focus solely on variations in loss, gradient, or parameters, neglecting the "bias" phenomenon caused by differing client dominance due to varying data volumes. Since the label shift scenario involves highly imbalanced client data volumes, both FeSEM and CFL-GP exhibit large NGP metrics, signifying the presence of severe negative gains.

Furthermore, regarding the high-quality distribution shift distance disparity we consider. Except for the loss-based

*Table 3.* Ablation study on the impact of distribution shift and quantity skew.

| Method | CIFAR10 (Feature Shift) | | | FMNIST (Label Shift) | | | CIFAR100 (Quality Shift) | | |
|---|---|---|---|---|---|---|---|---|---|
| | ACC↑ | NGP↓ | RSD↓ | ACC↑ | NGP↓ | RSD↓ | ACC↑ | NGP↓ | RSD↓ |
| Only Distribution | 44.12 (0.84) | 5.00 (2.50) | 3.76 (0.62) | 52.26 (1.45) | 60.00 (5.00) | 34.52 (2.10) | 26.57 (0.76) | 60.00 (5.00) | 11.02 (1.34) |
| Only Quantity | 47.84 (0.92) | 0.00 (0.00) | 2.99 (0.41) | 86.03 (0.68) | 10.00 (2.50) | 5.76 (0.95) | 34.44 (0.88) | 10.00 (5.00) | 6.41 (1.02) |
| **FedGain** | **50.44** (0.36) | **0.00** (0.00) | 3.20 (0.64) | **90.16** (0.59) | **0.00** (0.00) | 5.29 (1.13) | **37.95** (0.82) | **0.00** (0.00) | 7.32 (2.69) |

*Table 4.* Performance Comparison on OfficeHome Dataset.

| Method | OfficeHome (Domain Shift) | | |
|---|---|---|---|
| | ACC↑ | NGP↓ | RSD↓ |
| Local Train | 62.09 (1.52) | - | - |
| FedAvg | 65.42 (0.45) | 25.00 (2.50) | 5.81 (0.55) |
| **+FedGain** | **77.22** (0.68) | **0.00** (0.00) | **4.78** (0.42) |
| FedProx | 65.46 (0.58) | 30.00 (5.00) | 6.02 (0.25) |
| **+FedGain** | **76.95** (0.85) | **0.00** (0.00) | **4.87** (0.38) |
| FedNova | 65.45 (0.48) | 20.00 (0.00) | 5.86 (1.02) |
| **+FedGain** | **77.17** (0.72) | **0.00** (0.00) | **5.08** (0.45) |
| FedCSD | 56.95 (0.65) | 50.00 (5.00) | 5.69 (0.68) |
| **+FedGain** | **76.95** (1.10) | **0.00** (0.00) | **4.47** (0.35) |
| Finetune | 65.55 (0.88) | 21.00 (2.00) | 5.78 (0.42) |
| **+FedGain** | **77.04** (0.95) | **0.00** (0.00) | **4.94** (0.50) |
| Per-FedAvg | 56.74 (3.12) | 34.00 (2.00) | 5.33 (0.75) |
| **+FedGain** | **77.04** (1.05) | **0.00** (0.00) | **4.94** (0.48) |
| pFedMe | 48.13 (3.45) | 58.00 (2.45) | 9.90 (2.80) |
| **+FedGain** | **71.01** (1.28) | **0.00** (0.00) | **8.73** (1.92) |
| Ditto | 65.73 (0.74) | 25.00 (5.00) | 5.79 (0.28) |
| **+FedGain** | **77.44** (0.82) | **0.00** (0.00) | **5.11** (0.62) |
| FedAS | 66.11 (0.55) | 10.00 (0.00) | 5.10 (0.34) |
| **+FedGain** | **77.55** (0.48) | **0.00** (0.00) | **5.26** (0.18) |
| IFCA | 49.38 (0.92) | 28.00 (2.45) | 6.12 (0.57) |
| FedCluster | 46.64 (1.18) | 45.00 (5.00) | 7.45 (0.88) |
| FeSEM | 47.02 (1.30) | 27.50 (2.50) | 6.89 (0.95) |
| CFL-GP | 46.14 (1.05) | 34.00 (2.00) | 7.13 (1.11) |

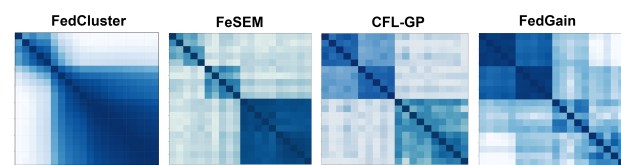

*Figure 3.* Client Distance Matrix for Cifar10 Feature Shift Scenario.

OfficeHome dataset.

OfficeHome consists of four natural domains (Art, Clipart, Product, and Real World), where inter-domain discrepancies stem from actual variations in equipment and environments. In federated learning research, it is widely utilized to assess a model's capability in handling cross-domain heterogeneity. The significant differences between these domains simulate the natural distribution shifts in cross-silo scenarios, which typically arise from the diverse acquisition devices and environments used by different organizations.

The 65 categories and cross-domain distribution mismatch provided by OfficeHome create a rigorous testing environment for validating FedGain's accuracy in optimizing collaboration structures within high-dimensional manifold spaces.

Table 4 reports the comparative results between FedGain and various baselines on the OfficeHome task. Even under complex domain-shift scenarios, FedGain maintains a NGP of 0%. This provides strong evidence that our structure optimization mechanism can precisely identify and exclude harmful collaborations caused by severe distribution shifts. Consequently, it ensures that every participating client achieves a net benefit relative to local training, even within complex and heterogeneous environments.

### 5.5. Ablation Study

We perform ablation studies to evaluate the synergistic effects of distribution shift and quantity skew in optimizing client collaboration. To decouple these factors, we alternatively nullify one error term while maintaining the other during computation.

Table 3 presents the results of the ablation study. We observe that the performance of the only distribution shift method degrades significantly across different distribution scenarios. This leads to a phenomenon where clients with small data volumes and those with large data volumes are grouped into two separate clusters, causing the model to become biased

IFCA method, the other approaches all involve distance measurement. Figure 3 presents the distance-based clustering results. All four methods exhibit clear "block-diagonal" patterns, indicating their effectiveness in clustering clients with identical distributions. However, according to the aforementioned feature shift scenario setup, the FedGain method most clearly reveals the characteristic that clients are divided into four distinct shift groups.

It is noteworthy that the clustering in baseline methods is performed multiple times during the FL training process, whereas the clustering in the FedGain method occurs prior to FL training. Consequently, it can be effectively integrated with other GFL or PFL methods, while imposing minimal impact on the overall overhead.

### 5.4. Cross-Silo Experiment Analysis

To further verify the robustness of FedGain in complex real-world scenarios, we conduct a evaluation using the

toward the cluster with larger data volume. Similarly, the performance of the only quantity skew method also declines notably. The performance degradation is more pronounced when the distribution shift is more significant, such as when clients with completely non-overlapping labels are assigned to the same cluster.

### 5.6. Quantity-Aware Scaling Curves

We evaluate the standard scaling law (SL) and our proposed heterogeneity-aware SL by constructing scaling curves across IID and non-IID federated scenarios. As illustrated in Figure 4-A for CIFAR-100, after recalibrating non-IID data using $M_{eff}$, the scaling curves for both settings exhibit highly parallel power-law trajectories. Notably, while data heterogeneity induces a performance lag due to $M_{eff} < M_{total}$, it does not alter the fundamental learning dynamics, as evidenced by the consistent slope. Essentially, heterogeneity acts as an offset that diminishes effective data value, shifting the scaling origin without compromising the underlying power-law discipline.

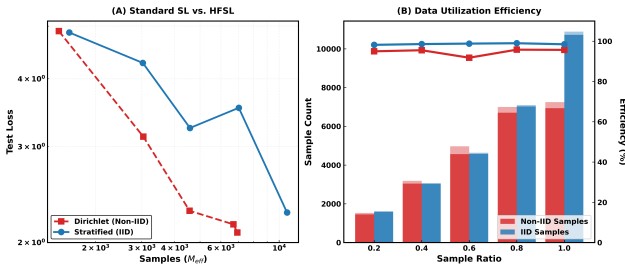

*Figure 4.* Scaling Curves and Data Quantity Utilization.

Figure 4-B demonstrates a significant drop in the utilization rate of non-IID data affected by distribution shift, where the effective sample size $M_{eff}$ participating in meaningful federated collaboration decreases. In the IID environment, due to consistent distributions across clients, data utilization efficiency approaches 100%.

### 5.7. Hyperparameter Analysis

In experiments on the dynamic calibration of scaling exponent $\nu$, the observed decay of $\nu$ as sample size $m$ increases aligns with the principle of diminishing marginal utility in deep learning. This strongly supports our shift from empirical fixation to data-driven calibration. Furthermore, once $\nu$ captures the core nonlinear dynamics, parameters $C$ and $\gamma$ exhibit remarkable stability across various scales.

The hyperparameter $C$ to modulate the model's sensitivity to sample volume. Increasing $C$ numerically amplifies the marginal utility of effective sample increments on test loss reduction. Empirically, $C \in \{8, 10\}$ yields optimal and identical collaboration structures, achieving the peak

accuracy gain. Notably, the framework degenerates to local training as $C \to 0$, while asymptotically converging to global training as $C \to \infty$.

Furthermore, the sensitivity parameter $\gamma$ in our decaying kernel function dictates the decay rate relative to distribution shift distance. In our unified experiments, $\gamma = 5.0$ provides the most significant accuracy improvement. Mechanistically, $M_{eff}$ reduces to $M_{total}$ when $\gamma = 0$, whereas an excessively large $\gamma$ causes $M_{eff}$ to vanish, leading to training divergence. The relevant experiments are shown in Appendix B.3.

## 6. Conclusion

This paper proposes FedGain, a clustering-guided framework for optimizing client collaboration relationships. It introduces feature contrast to quantify distribution distance, adapts SL for federated scenarios, and determines client collaboration through the deep coupling and optimization of distribution shift and quantity skew information, thereby achieving the elimination of negative gains. Experiments demonstrate that the scaling curves exhibit highly parallel power-law trends. Moreover, when combined with either GFL or PFL methods, the framework yields significant performance improvements, achieving higher accuracy and better cluster structure compared to CFL approaches.

## Acknowledgements

This work was supported in part by National Natural Science Foundation of China under Grant No. (62572202, 61972168, 61802134), the National Key Research and Development Program of China under Grant No. 2023YFC3304501, the Fundamental Research Funds for the Central Universities (ZQN-811), the Natural Science Foundation of Fujian Province of China under Grant No. 20231ZB038.

## Impact Statement

This paper presents work whose goal is to advance the field of Machine Learning. There are many potential societal consequences of our work, none which we feel must be specifically highlighted here.

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

# A. Proofs

## A.1. Proof of Theorem 3.3

We provide the formal proof of Theorem 3.3. The definitions of the distribution shift error term $D(P_i, P_j)$ and the data-dependent generalization error term $\Psi(\alpha, m, \mathcal{D}, \delta)$ are provided in the main text.

### A.1.1. DERIVATION OF THE DISTRIBUTION SHIFT ERROR $D(P_i, P_j)$

**Definition 3.1** Given a feature space $\mathcal{H}$, the risk upper bound between two distributions $P_i$ and $P_j$ induced by the distribution shift is defined as:

$$|r_i(h) - r_j(h)| \le L \cdot D(P_i, P_j) \tag{15}$$

where $L$ is the Lipschitz constant associated with the hypothesis space $\mathcal{H}$.

To ensure the theoretical validity of the risk bounds in the feature space, we introduce the following standard assumption regarding the feature extractor and the loss function.

**Assumption 1** (Regularity of Feature Mapping). The feature extractor $\phi : \mathcal{X} \to \mathcal{Z}$ is $L_\phi$-Lipschitz continuous. Specifically, for any $x, x' \in \mathcal{X}$, $\|\phi(x) - \phi(x')\|_2 \le L_\phi \|x - x'\|_2$. Furthermore, the feature space $\mathcal{Z}$ is compact and lies on a unit hypersphere (as enforced by the normalization in contrastive learning).

**Assumption 2** (Smoothness of Loss). The loss function $\ell(\cdot, \cdot)$ is $L_{loss}$-Lipschitz continuous and $\beta$-smooth with respect to the model output in the feature space.

**Lemma 1** ($L$-Lipschitz Continuity in Feature Space $\mathcal{H}$). Based on Assumptions 1 and 2, the feature extractor $\phi$ is Lipschitz continuous and the loss function $\ell$ is $L$-Lipschitz continuous with respect to the model output, then the difference in expected risk between clients is $L$-Lipschitz continuous with respect to the distribution shift distance $D(P_i, P_j)$.

*Proofs.*

Let $h \in \mathcal{H}$ be a hypothesis defined on the feature space. By definition, the upper bound on the expected risk difference is:

$$|r_i(h) - r_j(h)| = \left| \int \ell(h(x), y) dP_i(x) - \int \ell(h(x), y) dP_j(x) \right| \tag{16}$$

In the mapped feature space $Z = \phi(x)$, let $L_{loss}$ be the Lipschitz constant of $\ell$ with respect to the feature $z$. It follows that:

$$|r_i(h) - r_j(h)| \le L_{loss} \cdot \|\mathbb{E}_{x \sim P_i}[\phi(x)] - \mathbb{E}_{x \sim P_j}[\phi(x)]\|_2 \tag{17}$$

According to **Definition 3.1**, the distribution shift $D(P_i, P_j)$ is defined as the normalized cosine distance between the feature centroids $\mu_i$ and $\mu_j$. On the unit sphere, the Euclidean distance is equivalent to the cosine distance. There exists a constant $L = \sqrt{2} \cdot L_{loss}$ such that:

$$|r_i(h) - r_j(h)| \le L \cdot D(P_i, P_j) \tag{18}$$

This completes the proof that $D(P_i, P_j)$ directly constrains the deviation of the risk function.

**Lemma 2** (Bias Bound via $\mathcal{H}\Delta\mathcal{H}$-divergence). Given the cluster mixture distribution $P_{\mathcal{C}_k} = \sum_{j \in \mathcal{C}_k} \alpha_j P_j$, the risk bias between the target client $i$ and the cluster distribution is bounded by the weighted sum of the distances between member distributions.

*Proofs.*

According to domain adaptation theory (Ben-David et al., 2010), the $\mathcal{H}\Delta\mathcal{H}$-divergence measures the generalization deviation of a model across different distributions. In the feature space aligned by our contrastive learning approach, this divergence is characterized by $D(P_i, P_j)$. Utilizing the linearity of the expected risk:

$$|r_i(h) - r_{\mathcal{C}_k}(h)| = \left| r_i(h) - \sum_{j \in \mathcal{C}_k} \alpha_j r_j(h) \right| = \left| \sum_{j \in \mathcal{C}_k} \alpha_j (r_i(h) - r_j(h)) \right| \tag{19}$$

By applying the conclusion of Lemma 1 and the triangle inequality:

$$|r_i(h) - r_{\mathcal{C}_k}(h)| \leq \sum_{j \in \mathcal{C}_k} \alpha_j |r_i(h) - r_j(h)| \leq \sum_{j \in \mathcal{C}_k, j \neq i} \alpha_j D(P_i, P_j) \tag{20}$$

This term corresponds to the distribution shift error in **Theorem 3.3**, describing the error introduced by distribution inconsistencies among collaborative participants.

A.1.2. DERIVATION OF THE GENERALIZATION ERROR $\Psi(\alpha, m, D, \delta)$

**Definition 3.2** For any client $i$ in cluster $\mathcal{C}_k$, given a confidence parameter $\delta \in (0, 1)$, there exist a scaling exponent $\nu > 0$ and a constant $C$ such that the generalization error $\Psi$ satisfies the following with probability at least $1 - \delta$:

$$|r_{\mathcal{C}_k}(h) - \hat{r}_{\mathcal{C}_k}(h)| \leq \Psi(\alpha, m, D, \delta) \tag{21}$$

In the main text, we define:

$$\Psi_i(\alpha, m, D, \delta) = C \left( m_i + \sum_{j \in \mathcal{C}_k, j \neq i} \alpha_j \cdot m_j \cdot e^{-\gamma D(P_i, P_j)} \right)^{-\nu} \tag{22}$$

**Proposition 1** (Heterogeneity-Aware Convergence). Within cluster $\mathcal{C}_k$, assume the data from all participants $j \in \mathcal{C}_k$ contributes information via weighted aggregation for a target client $i$. Under $1 - \delta$ confidence, the generalization error bound satisfies the power-law relationship from the Scaling Laws (Kaplan et al., 2020):

$$\Psi_i \propto \left( m_i + \sum_{j \neq i} m_j \cdot \beta(D(P_i, P_j)) \right)^{-\nu} \tag{23}$$

where $\beta(D(P_i, P_j)) = e^{-\gamma D(P_i, P_j)}$ represents the heterogeneity attenuation factor.

*Analysis.*

In federated aggregation, the existence of distribution shift $D(P_i, P_j)$ renders samples from client $j$ "biased" relative to the target distribution $P_i$. According to Importance Sampling theory (Gretton et al., 2009), when using samples from $P_j$ to estimate the risk on $P_i$, the estimator variance expands due to the distributional distance.

Based on Optimal Weighted Estimation theory, a dataset with size $m_j$ and distribution distance $D(P_i, P_j)$ provides a Statistical Utility equivalent to $m_j \cdot W_{ij}$ IID samples. We define the weight $W_{ij} = \beta(D(P_i, P_j))$. As $D(P_i, P_j) \to 0$ (distributional consistency), $W_{ij} \to 1$; as $D(P_i, P_j) \to \infty$, $W_{ij} \to 0$.

Let $\ell(h; x)$ be the loss function. Our goal is to estimate the risk $r_i(h) = \mathbb{E}_{x \sim P_i}[\ell(h; x)]$ using samples from $P_j$. By Importance Sampling:

$$r_i(h) = \mathbb{E}_{x \sim P_j} \left[ \frac{P_i(x)}{P_j(x)} \ell(h; x) \right] \tag{24}$$

Setting the weight $\omega(x) = \frac{P_i(x)}{P_j(x)}$, the variance of this estimator is:

$$\text{Var}_{P_j}[\hat{r}_i] = \frac{1}{m_j} \text{Var}_{P_j}[\omega(x)\ell(h; x)] \approx \frac{1}{m_j} \mathbb{E}_{P_j}[\omega(x)^2] \cdot \sigma^2 \tag{25}$$

Due to the positive correlation between the $\chi^2$-divergence and the distribution shift distance $D(P_i, P_j)$, $\mathbb{E}[\omega(x)^2]$ grows exponentially as $D(P_i, P_j)$ increases. This implies that to achieve the same estimation precision (variance) as IID samples, the utility of non-IID samples is "diluted."

**Supplementary:Derivation of the Exponential Decay Factor**

To rigorously justify the exponential relationship $\mathbb{E}_{P_j}[\omega(x)^2] \propto e^{\gamma D(P_i, P_j)}$, we analyze the feature distributions in the contrastive-aligned space.

**Assumption 3** (vMF Distribution on Hypersphere). Following standard representation learning theory, we assume the normalized feature representations $z = \phi(x)$ for class $y$ on client $i$ follow a Von Mises-Fisher (vMF) distribution: $P_i(z) = C_d(\kappa)e^{\kappa \mu_i^\top z}$, where $\mu_i$ is the distribution centroid and $\kappa$ is the concentration parameter.

The importance weight second moment is given by:

$$
\begin{aligned}
\mathbb{E}_{P_j}\left[\left(\frac{P_i(z)}{P_j(z)}\right)^2\right] &= \int P_j(z)\frac{P_i(z)^2}{P_j(z)^2}dz \\
&= \int \frac{(C_d(\kappa)e^{\kappa \mu_i^\top z})^2}{C_d(\kappa)e^{\kappa \mu_j^\top z}}dz \\
&= \frac{C_d(\kappa)^2}{C_d(\kappa)} \int \frac{e^{2\kappa \mu_i^\top z}}{e^{\kappa \mu_j^\top z}}dz \\
&= C_d(\kappa) \int \exp\left(\kappa(2\mu_i - \mu_j)^\top z\right) dz
\end{aligned}
\tag{26}
$$

This integral corresponds to the partition function of a new vMF distribution with direction $2\mu_i - \mu_j$. The value of this integral scales exponentially with the length of the vector $\|2\mu_i - \mu_j\|$.

Noting that $D(P_i, P_j) \approx \|\mu_i - \mu_j\|^2$ (on the unit sphere), the term can be approximated as:

$$
\mathbb{E}_{P_j}[\omega(z)^2] \approx \exp\left(\gamma\|\mu_i - \mu_j\|^2\right) = \exp(\gamma D(P_i, P_j))
\tag{27}
$$

Thus, the effective sample size is scaled by the inverse of this variance:

$$
\beta(D(P_i, P_j)) = \frac{1}{\mathbb{E}[\omega(z)^2]} \approx e^{-\gamma D(P_i, P_j)}
\tag{28}
$$

This derivation confirms that under the vMF assumption inherent to contrastive feature spaces, the attenuation factor strictly follows an exponential decay.

Following the Effective Sample Size (ESS) formula for heterogeneous observations (Martino et al., 2017) and the variance expansion factor $\mathbb{E}_{P_j}[\omega(x)^2]$, the statistical contribution of samples from client $j$ to client $i$ is defined as:

$$
m_j^{\text{eff}} = \frac{m_j}{\mathbb{E}_{P_j}[\omega(x)^2]} \triangleq m_j \cdot \beta(D(P_i, P_j))
\tag{29}
$$

In the feature-aligned space, given the properties of the exponential family distributions, the attenuation kernel $\beta(\cdot)$ can be strictly derived as the exponential form $\beta(D(P_i, P_j)) = e^{-\gamma D(P_i, P_j)}$. Thus, the Effective Federated Capacity of cluster $\mathcal{C}_k$ for client $i$ is:

$$
M_{\text{eff}}(i) = m_i + \sum_{j \neq i} m_j \cdot \beta(D(P_i, P_j))
\tag{30}
$$

*Proofs.*

Since the data is non-IID, the generalization performance of the cluster model $h_{\mathcal{C}_k}$ depends on the statistical effective sample

size.

$$
\begin{aligned}
|r_{\mathcal{C}_k}(h) - \hat{r}_{\mathcal{C}_k}(h)| &\leq \mathcal{R}_{M_{\text{total}}}(\mathcal{H}) + \sqrt{\frac{\log(1/\delta)}{2M_{\text{total}}}} && \text{(Classical Rademacher complexity)} \\
&\approx C \cdot [M_{\text{eff}}^{(i)}]^{-\frac{1}{2}} && \text{(Statistical rate via CLT)} \\
&\leq C \cdot [M_{\text{eff}}^{(i)}]^{-\nu} && \text{(Kaplan et al., 2020)} \\
&= C \cdot \left[ m_i + \sum_{j \in \mathcal{C}_k, j \neq i} \text{Utility}(m_j, D(P_i, P_j)) \right]^{-\nu} \\
&= C \cdot \left[ m_i + \sum_{j \neq i} \text{ESS}(m_j) \right]^{-\nu} \\
&\leq C \cdot \left[ m_i + \sum_{j \neq i} m_j \cdot \mathbb{E}_{P_j}\left[ \frac{P_i(x)}{P_j(x)} \right]^{-1} \right]^{-\nu} \\
&\approx C \cdot \left[ m_i + \sum_{j \neq i} m_j \cdot e^{-\gamma D(P_i, P_j)} \right]^{-\nu} \\
&\leq C \cdot \left[ m_i + \sum_{j \in \mathcal{C}_k, j \neq i} \alpha_j \cdot m_j \cdot e^{-\gamma D(P_i, P_j)} \right]^{-\nu} && \text{(Introducing cluster weights)} \\
&= \Psi_i(\alpha, m, D, \delta)
\end{aligned}
\tag{31}
$$

This term corresponds to the data-dependent generalization error in Theorem 3.3, characterizing the statistical instability of the model under finite observations. Specifically, it describes the information dilution effect during the aggregation of heterogeneous data. Unlike traditional generalization bounds, our $\Psi_i$ utilizes $M_{\text{eff}}$ to distinguish "effective augmentation" from "ineffective accumulation," providing theoretical guidance for clustering structure optimization at the data-scale level.

### A.1.3. ERROR UPPER BOUND

**Theorem 3.3** Within cluster $\mathcal{C}_k$, let $h_i^*$ be the theoretical optimal model for client $i$. Combining Definitions 3.1 and 3.2, for any $h \in \mathcal{H}$, the expected risk of client $i$ using the cluster model $h_{\mathcal{C}_k}$ satisfies the following upper bound with probability at least $1 - \delta$:

$$
r_i(h_{\mathcal{C}_k}) \leq r_i(h_i^*) + 2 \sum_{j \in C_k, j \neq i} \alpha_j D(P_i, P_j) + 2\Psi_i(\alpha, m, D, \delta)
\tag{32}
$$

*Proofs.*

$$
\begin{aligned}
r_i(h_{\mathcal{C}_k}) &\leq r_i(h_i^*) + |r_i(h_{\mathcal{C}_k}) - r_i(h_i^*)| \\
&\leq r_i(h_i^*) + |r_i(h_{\mathcal{C}_k}) - r_{\mathcal{C}_k}(h_{\mathcal{C}_k})| + |r_{\mathcal{C}_k}(h_{\mathcal{C}_k}) - r_{\mathcal{C}_k}(h_i^*)| + |r_{\mathcal{C}_k}(h_i^*) - r_i(h_i^*)| \\
&\leq r_i(h_i^*) + \sum_{j \in \mathcal{C}_k} \alpha_j |r_i(h_{\mathcal{C}_k}) - r_j(h_{\mathcal{C}_k})| + |r_{\mathcal{C}_k}(h_{\mathcal{C}_k}) - r_{\mathcal{C}_k}(h_i^*)| + \sum_{j \in \mathcal{C}_k} \alpha_j |r_j(h_i^*) - r_i(h_i^*)| \\
&\leq r_i(h_i^*) + \sum_{j \in \mathcal{C}_k, j \neq i} \alpha_j D(P_i, P_j) + |r_{\mathcal{C}_k}(h_{\mathcal{C}_k}) - r_{\mathcal{C}_k}(h_i^*)| + \sum_{j \in \mathcal{C}_k, j \neq i} \alpha_j D(P_i, P_j) && \text{(Theorem 3.1)} \\
&= r_i(h_i^*) + 2 \sum_{j \in \mathcal{C}_k, j \neq i} \alpha_j D(P_i, P_j) + |r_{\mathcal{C}_k}(h_{\mathcal{C}_k}) - r_{\mathcal{C}_k}(h_i^*)| \\
&\leq r_i(h_i^*) + 2 \sum_{j \in \mathcal{C}_k, j \neq i} \alpha_j D(P_i, P_j) + |r_{\mathcal{C}_k}(h_{\mathcal{C}_k}) - \hat{r}_{\mathcal{C}_k}(h_{\mathcal{C}_k})| + |\hat{r}_{\mathcal{C}_k}(h_{\mathcal{C}_k}) - \hat{r}_{\mathcal{C}_k}(h_i^*)| + |\hat{r}_{\mathcal{C}_k}(h_i^*) - r_{\mathcal{C}_k}(h_i^*)| \\
&\leq r_i(h_i^*) + 2 \sum_{j \in \mathcal{C}_k, j \neq i} \alpha_j D(P_i, P_j) + \Psi_i(\alpha, m, D, \delta) + |\hat{r}_{\mathcal{C}_k}(h_{\mathcal{C}_k}) - \hat{r}_{\mathcal{C}_k}(h_i^*)| + \Psi_i(\alpha, m, D, \delta) && \text{(Theorem 3.2)} \\
&\leq r_i(h_i^*) + 2 \sum_{j \in \mathcal{C}_k, j \neq i} \alpha_j D(P_i, P_j) + 2\Psi_i(\alpha, m, D, \delta) + 0 && \text{(By ERM.)} \\
&= r_i(h_i^*) + 2 \sum_{j \in \mathcal{C}_k, j \neq i} \alpha_j D(P_i, P_j) + 2\Psi_i(\alpha, m, D, \delta)
\end{aligned}
$$
$$\tag{33}$$

Since the terms decomposed via the triangle inequality each hold with probability $1 - \delta$, the final inequality holds with probability at least $1 - \delta$ according to the Union Bound. Furthermore, based on the fundamental properties of Empirical Risk Minimization (ERM), since the cluster model $h_{\mathcal{C}_k}$ is obtained by optimizing over the clustered dataset $\mathcal{D}_{\mathcal{C}_k}$, its empirical loss on this dataset is necessarily less than or equal to that of any other model, including $h_i^*$ (i.e., $\hat{r}_{\mathcal{C}_k}(h_{\mathcal{C}_k}) \leq \hat{r}_{\mathcal{C}_k}(h_i^*)$).

## A.2. Proof of Corollary

**Corollary 3.4** (Condition for Positive Collaboration Gain) Let $\mathcal{R}_{local}^{(i)}$ be the generalization error bound of client $i$ training solely on its local dataset $\mathcal{D}_i$ of size $m_i$, and $\mathcal{R}_{fed}^{(i)}$ be the bound derived in **Theorem 3.3** for cluster-based training. A Positive Gain (i.e., $\mathcal{R}_{fed}^{(i)} < \mathcal{R}_{local}^{(i)}$) is guaranteed if the reduction in generalization error exceeds the induced distribution bias:

$$
\underbrace{\Psi_i(1, m_i, 0, \delta) - \Psi_i(\alpha, m, D, \delta)}_{\text{Gain from Effective Capacity}} > 2 \underbrace{\sum_{j \in \mathcal{C}_k, j \neq i} \alpha_j D(P_i, P_j)}_{\text{Cost of Distribution Shift}}
$$
$$\tag{34}$$

*Proofs.*

The risk bound for local training (where $D(P_i, P_i) = 0$ and only $m_i$ is used) follows the Scaling Law:

$$
r_i(h_{local}) \leq r_i(h^*) + C \cdot (m_i)^{-\nu}
$$
$$\tag{35}$$

The risk bound for clustered training (**Theorem 3.3**) is:

$$
r_i(h_{\mathcal{C}_k}) \leq r_i(h^*) + 2 \sum_{j \neq i} \alpha_j D(P_i, P_j) + C \cdot (M_{\text{eff}}(i))^{-\nu}
$$
$$\tag{36}$$

Subtracting the cluster bound from the local bound, the condition $r_i(h_{\mathcal{C}_k}) < r_i(h_{local})$ holds when:

$$
C \cdot (m_i)^{-\nu} - \left( 2 \sum_{j \neq i} \alpha_j D(P_i, P_j) + C \cdot (M_{\text{eff}}(i))^{-\nu} \right) > 0
$$
$$\tag{37}$$

Rearranging the terms yields the condition stated in the Corollary. This inequality theoretically validates that FedGain achieves positive gains by maximizing $M_{\text{eff}}(i)$ (via high-affinity client selection) while explicitly constraining the distribution shift term $\sum \alpha_j D(P_i, P_j)$.

### A.3. Convergence Proof of the Optimizer

**Convergence.** Suppose $\mathcal{L}_s$ is Lipschitz continuous and $L$-smooth on the probability simplex $\Delta_K$. If the learning rate $\eta < 1/L$, the sequence $\{S^{(t)}\}$ generated by Projected Gradient Descent (PGD) satisfies $\mathcal{L}_s(S^{(t+1)}) \leq \mathcal{L}_s(S^{(t)})$ and converges to a first-order stationary point.

*Proofs.*

Based on the definition of $L$-smoothness, the function value at $S^{(t+1)}$ is bounded by its first-order expansion at $S^{(t)}$:

$$\mathcal{L}(S^{(t+1)}) \leq \mathcal{L}(S^{(t)}) + \langle \nabla \mathcal{L}(S^{(t)}), S^{(t+1)} - S^{(t)} \rangle + \frac{L}{2} \|S^{(t+1)} - S^{(t)}\|^2 \tag{38}$$

The update rule for PGD is $S^{(t+1)} = \text{Proj}_\Delta(S^{(t)} - \eta \nabla \mathcal{L}(S^{(t)}))$. According to the optimality condition for projection onto a convex set, for any $y \in \Delta$, the following holds:

$$\langle S^{(t+1)} - (S^{(t)} - \eta \nabla \mathcal{L}(S^{(t)})), y - S^{(t+1)} \rangle \geq 0 \tag{39}$$

Setting $y = S^{(t)}$ (which is valid as $S^{(t)}$ lies within the simplex), substituting into Eq. (39) yields:

$$\langle S^{(t+1)} - S^{(t)} + \eta \nabla \mathcal{L}(S^{(t)}), S^{(t)} - S^{(t+1)} \rangle \geq 0 \tag{40}$$

Expanding and rearranging Eq. (40):

$$\eta \langle \nabla \mathcal{L}(S^{(t)}), S^{(t)} - S^{(t+1)} \rangle \geq \|S^{(t+1)} - S^{(t)}\|^2 \iff \langle \nabla \mathcal{L}(S^{(t)}), S^{(t+1)} - S^{(t)} \rangle \leq -\frac{1}{\eta} \|S^{(t+1)} - S^{(t)}\|^2 \tag{41}$$

Substituting Inequality Eq. (41) into the inner product term of Eq. (38) ensures that the direction of gradient descent is consistent with the displacement after projection:

$$\begin{aligned}
\mathcal{L}(S^{(t+1)}) &\leq \mathcal{L}(S^{(t)}) + \langle \nabla \mathcal{L}(S^{(t)}), S^{(t+1)} - S^{(t)} \rangle + \frac{L}{2} \|S^{(t+1)} - S^{(t)}\|^2 \\
&\leq \mathcal{L}(S^{(t)}) - \frac{1}{\eta} \|S^{(t+1)} - S^{(t)}\|^2 + \frac{L}{2} \|S^{(t+1)} - S^{(t)}\|^2 \\
&= \mathcal{L}(S^{(t)}) - \left( \frac{1}{\eta} - \frac{L}{2} \right) \|S^{(t+1)} - S^{(t)}\|^2
\end{aligned} \tag{42}$$

To ensure $\mathcal{L}(S^{(t+1)}) \leq \mathcal{L}(S^{(t)})$ (the condition for monotonic decrease), we require $\frac{1}{\eta} - \frac{L}{2} > 0 \Rightarrow \eta < \frac{2}{L}$. Summing from $t = 0$ to $T$:

$$\sum_{t=0}^{T} \left( \frac{1}{\eta} - \frac{L}{2} \right) \|S^{(t+1)} - S^{(t)}\|^2 \leq \mathcal{L}(S^{(0)}) - \mathcal{L}(S^{(T+1)}) \tag{43}$$

Since $\mathcal{L}$ is bounded from below (minimum risk $r \geq 0$), the right-hand side remains bounded as $T \to \infty$, which implies:

$$\lim_{t \to \infty} \|S^{(t+1)} - S^{(t)}\| = 0 \tag{44}$$

In Eq. (38), the term $e^{-\gamma D_{ij}}$ contained within the gradient $\nabla \mathcal{L}$ ensures that: when distributions are similar ($D_{ij}$ is small, $e^{-\gamma D_{ij}} \to 1$), the gradient is large, creating a strong "attraction" that forces the algorithm to increase the value of $s_{ik}$, encouraging client $i$ to join the cluster. Conversely, when the distribution shift is extreme (large $D_{ij}$), the exponential decay effect causes $e^{-\gamma D_{ij}}$ to rapidly approach 0, and the contribution of this term to the gradient nearly vanishes. This implies that if two distributions are entirely different, the gradient will be suppressed regardless of the data volume ($m_j$).

The algorithm ignores such cases, thereby avoiding the introduction of significant distribution bias (model drift) caused by blindly pursuing large data volumes. This allows $S$ to eventually stabilize at the edges of the simplex (at discrete solutions), enabling the algorithm to robustly identify clear clustering structures.

**Stability Analysis: Sparsity Inducement via Simplex Projection**

Finally, we analyze why the sequence $\{S^{(t)}\}$ converges to a definitive clustering structure (where $s_{ik} \to 0$ or $1$) rather than a soft ambiguity. The PGD update step involves a projection onto the probability simplex $\Delta_K$:

$$S^{(t+1)} = \text{Proj}_\Delta(S^{(t)} - \eta\nabla\mathcal{L}) \tag{45}$$

The simplex constraint $\sum_k s_{ik} = 1$ introduces a competitive mechanism.

Winner-Take-All Dynamics: As discussed in Eq. (44), the gradient term $e^{-\gamma D_{ik}}$ acts as an affinity score. If client $i$ is slightly closer to cluster $k$ (smaller $D_{ik}$), the gradient magnitude for component $k$ will be significantly larger than for other clusters.

Projection Sparsity: The Euclidean projection onto the simplex is known to promote sparsity. When one component $s_{ik}$ increases due to a strong gradient, the projection operator forces other components $s_{ij}(j \neq k)$ to shrink to satisfy the sum-to-one constraint. Once a component hits the boundary (0), the gradient update cannot make it negative, effectively locking it at 0 unless a significant distribution shift occurs.

Consequently, the combination of the exponentially sensitive gradient (which amplifies differences in $D_{ik}$) and the simplex projection (which enforces competition) drives the assignment vector $S_i$ toward the vertices of the simplex. This guarantees that FedGain converges to a stable, hard clustering structure, minimizing the ambiguity in client collaboration.

# B. Additional Experiments

### B.1. Validation of Key Assumptions (vMF)

In the preceding theoretical analysis, we assume that the decay kernel function is valid. Specifically, the framework assumes that the contribution of client $j$ to target client $i$ can be characterized by a monotonically decreasing kernel function $e^{-\gamma D}$. In highly heterogeneous real-world federated learning scenarios, the validity of this assumption depends on whether the feature distributions satisfy specific geometric constraints. Therefore, we propose that the features after the contrastive learning phase should follow a von Mises-Fisher (vMF) distribution, which serves as a critical theoretical prerequisite for deriving the exponential decay kernel.

As illustrated in Figure 5, we empirically validate the vMF assumption using the Mean Resultant Length ($R \in [0,1]$) on a unit hypersphere. Across the CIFAR-10/100 and FMNIST datasets, the $R$ values consistently stabilize around 0.9, indicating a high degree of directional concentration in the features. Notably, on the OfficeHome dataset, which contains natural domain shifts, the $R$ value remains robustly at 0.7832. Although slightly lower than those of synthetic datasets, $R \approx 0.78$ still represents significant cluster alignment, far exceeding the level of a random distribution. These experimental results confirm that our contrastive learning phase effectively maintains vMF-like feature structures even under substantial real-world heterogeneity, providing solid empirical support for our theoretical framework.

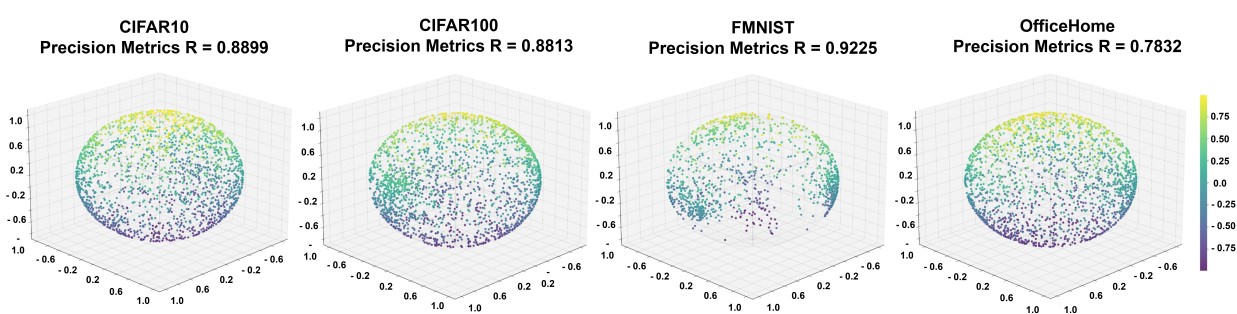

*Figure 5.* vMF hypothesis validation: visualizing data distribution on a hypersphere.

## B.2. Overhead and Scalability Analysis

We provide the end-to-end computation and communication overhead of FedGain compared to baseline methods. By monitoring the relative rate of change in the distribution shift matrix for 20 clients on the CIFAR-10 dataset, we observed that the comparison rounds stabilize at $T_c = 100$. Accordingly, we set the CFL methods (as GFL and PFL do not involve client group training) to perform 100 rounds of training to optimize their clustering structures. As shown in Table 5, FedGain outperforms the baselines in both communication overhead per round and average training time per client per round.

*Table 5.* End-to-End Compute and Communication Overhead.

| Method | Add. Rounds (Struct. Opt.) | Transmitted Size (Up/Down) [KB] | Avg. Local Time [s] |
|---|---|---|---|
| IFCA | 100 | 478.78 / 478.78$\times K$ | 0 |
| FedCluster | 100 | 478.78 / 478.78 | 0 |
| FeSEM | 100 | 478.78 / 478.78$\times K$ | 0 |
| CFL-GP | 100 | 478.78 / 478.78 | 0 |
| **FedGain** | **100** | **0.50 / 0** | **0.0873** |

In our method, each client uploads a feature projection vector to the server. With a dimension of 128 and a float32 data type (4 Bytes), each client transmits a fixed 0.5 KB per round. In contrast, the baseline methods transmit full models (either 1 or $K$ models), where a CNN model under float32 precision has a total size of 478.78 KB. Furthermore, FedGain performs preprocessing before the FL training begins. Instead of transmitting models to clients, it only sends instructions regarding cluster assignments (clustering indices), the overhead of which is negligible.

Additionally, Table 6 presents the total overhead of FedGain under different client scales. The server-side overhead is calculated as:

*Total Pipeline Execution Time (s) - (Average Local Compute Time × Rounds × N) - Fixed System Overhead = Matrix Computation Overhead + Structure Optimization Overhead.*

*Table 6.* Scalability Analysis of FedGain.

| Scale ($N$) | Avg. Local Time [s] | Total Exec. Time [s] | Server Overhead [s] |
|---|---|---|---|
| 20 | 0.0872 | 533.74 | 359.23 |
| 100 | 0.0308 | 684.53 | 376.50 |
| 1000 | 0.0043 | 1119.20 | 687.10 |

For an individual client, the overhead for feature projection computation and the 0.50 KB vector transmission is entirely independent of the total number of clients $N$. In a practical distributed system, all clients perform projection operations in parallel. Consequently, whether $N = 20$ or $N = 1000$, the computational load on each client remains extremely low. While the theoretical complexity of computing an $N \times N$ distance matrix is $O(N^2)$, distance calculations between such low-dimensional vectors (e.g., 128 dimensions) are highly efficient on modern servers. Even for a scale of $N = 1000$ involving $10^6$ distance operations, the computation takes only milliseconds when driven by efficient matrix operation libraries. Moreover, the complexity of the PGD algorithm used for cluster structure optimization primarily depends on the scale of the optimization variables (i.e., cluster association weights), and it demonstrates excellent convergence efficiency in practice.

Therefore, FedGain exhibits significant advantages in large-scale heterogeneous scenarios. First, regarding computational efficiency, as the client scale $N$ expands from 20 to 1000 (a 50-fold increase), the server-side optimization overhead increases by less than 100%, demonstrating strong sub-linear growth characteristics. Under the extreme pressure of $N = 1000$, the complex cluster optimization is completed in approximately 11 minutes. Second, in terms of communication scalability, FedGain achieves a balance of "high-frequency computation and minimalist communication." During the structure optimization phase, clients only need to upload a fixed 0.5 KB of metadata, and the downlink overhead is near zero (only cluster index assignments). This means the communication cost of FedGain is decoupled from the model parameter size, and its communication advantage scales linearly with $N$. The subsequent training overhead is primarily determined

by the integrated GFL or PFL algorithm. When combined with FedAvg for federated training, the total communication overhead remains at $O(\text{Model Size} \times N)$ without introducing additional complexity tiers due to clustering logic, fully proving its practicality in edge computing environments.

## B.3. Hyperparameter Analysis

In this section, we conduct systematic power-law fitting experiments ($L(m) \propto m^{-\nu}$) across multiple datasets. Supplementary experimental results regarding hyperparameters $\nu$, $C$, and $\gamma$ are provided.

### B.3.1. SCALING EXPONENT $\nu$

Figure 6 presents the results for the dynamic calibration of the scaling exponent $\nu$. The results indicate that $\nu$ exhibits a clear decaying trend as sample size $m$ increases. While decay baselines and rates vary by dataset, CIFAR10 consistently maintains a higher exponent, whereas FMNIST and CIFAR100 remain at lower levels ($\nu < 0.35$). Furthermore, the variance across the four heterogeneous groups remains stable, with the shaded regions demonstrating that the mean of $\nu$ is highly representative and the calibration process is robust across diverse data distributions.

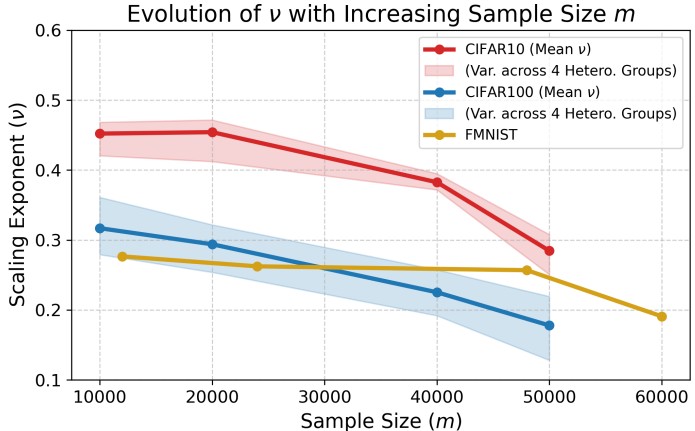

*Figure 6.* Evolution of Scaling Exponent with Increasing Sample Size.

### B.3.2. HYPERPARAMETER $C$

We evaluate FedGain by optimizing client collaboration for the FedAvg, with $\gamma$ fixed at 5.0. In the CIFAR-10 scenario, the resulting client collaboration structures remain identical for $C \in \{6.0, 8.0, 10.0\}$. In the FMNIST scenario, the structures are identical for $C = 6.0$ and $C = 8.0$; however, $C = 10.0$ yields a distinct collaboration structure that achieves the highest accuracy. For CIFAR-100, the collaboration structures for $C = 6.0$ and $C = 10.0$ are identical, while $C = 8.0$ produces a different structure with optimal accuracy performance.

*Table 7.* Performance comparison across different hyperparameters $C$ ($\gamma = 5.0$).

| Method / $C$ | CIFAR10 (Feature Shift) ACC↑ | NGP↓ | FMNIST (Label Shift) ACC↑ | NGP↓ | CIFAR100 (Quality Shift) ACC↑ | NGP↓ |
|---|---|---|---|---|---|---|
| Local Train | 35.63 (2.48) | - | 85.63 (0.07) | - | 29.18 (0.18) | - |
| FedAvg | 44.47 (0.27) | 10.00 (0.00) | 46.62 (0.06) | 55.00 (5.00) | 26.27 (0.05) | 50.00 (0.00) |
| 6.0 | 50.28 (0.84) | 0.00 (0.00) | 89.99 (0.71) | 10.00 (2.50) | 34.44 (0.76) | 10.00 (2.50) |
| 8.0 | 50.37 (0.92) | 0.00 (0.00) | 90.02 (0.68) | 10.00 (2.50) | **37.95 (0.82)** | **0.00 (0.00)** |
| 10.0 | **50.44 (0.36)** | **0.00 (0.00)** | **90.16 (0.59)** | **0.00 (0.00)** | 34.49 (0.86) | 10.00 (2.50) |

### B.3.3. HYPERPARAMETER $\gamma$

**Settings.** For the quantity-aware scaling experiments, we vary the sample utilization rates of CIFAR-100 data (0.2, 0.4, 0.6, 0.8, 1.0). Clients adopt varying sample utilization rates across five distinct experimental trials to simulate scenarios with different data scales. The heterogeneity settings in this scenario ensure that increasing the sample size simultaneously introduces a certain degree of heterogeneous data interference.

*Table 8.* Performance comparison under different hyperparameters $\gamma$ in the CIFAR100 scenario ($C = 8.0$).

| Method / $\gamma$ | ACC $\uparrow$ | NGP $\downarrow$ | IPR $\uparrow$ | RSD $\downarrow$ |
|---|---|---|---|---|
| Local Train | 29.18 (0.18) | - | - | - |
| FedAvg | 26.27 (0.05) | 50.00 (0.00) | 50.00 (0.00) | 12.56 (1.90) |
| 2.0 | 31.64 (0.65) | 35.00 (5.00) | 65.00 (5.00) | 7.74 (0.92) |
| 5.0 | **37.95 (0.82)** | **0.00 (0.00)** | **100.0 (0.00)** | **7.32 (2.69)** |
| 8.0 | 36.95 (0.78) | 10.00 (2.50) | 90.00 (2.50) | 6.12 (2.36) |

The loss function also incorporates the hyperparameter $\gamma$, which dictates the attenuation intensity of the kernel function relative to the distribution shift distance. We set $C = 8.0$ for the CIFAR-100 scenario. Table 8 displays the results of optimizing FedAvg through client collaboration. In this specific heterogeneous setting, the performance is maximized when $\gamma = 5.0$.

### B.3.4. HYPERPARAMETER $K$

Furthermore, the number of clusters $K$ resulting from client collaboration is indirectly influenced by the value of $\nu$. During the initialization phase, we set a reasonable upper bound for $K$ based on the client scale. When the solver minimizes the HFSL objective function, if the EFC gain from merging two clusters outweighs the penalty incurred by the distribution shift $D$, the optimization process naturally drives the membership weights toward a more compact structure. Figure 7 illustrates the impact of the scaling exponent $\nu$ on both the value of $K$ and cluster performance in the CIFAR-10 task.

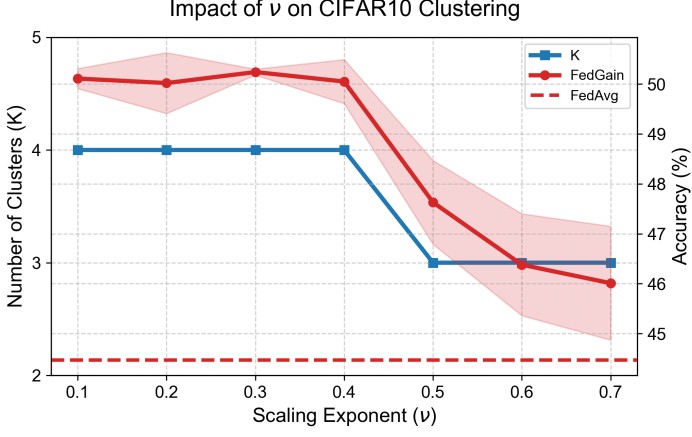

*Figure 7.* Impact of Scaling Exponent on Clustering Performance.

