# OpenReview forum: "FedGain: Toward Negative-Gain-Free Client Collaboration in Federated Learning"
_ICML.cc/2026/Conference — ICML 2026 regular_

### Official Review · Reviewer_9F2p · 2026-02-25

**Soundness:** 2
**Presentation:** 2
**Significance:** 3
**Originality:** 3
**Overall Recommendation:** 4
**Confidence:** 3

**Summary:**

This paper examines collaborative learning with heterogeneous data in federated learning (FL), focusing on the surprising phenomenon of negative gains, where some clients perform worse than when they train locally. The authors investigate why adding more data in FL does not always lead to better global results. They also aim to develop a collaboration method that prevents negative transfer by accounting for both distribution differences and data imbalances across clients. To tackle this, the paper presents a scaling framework that accounts for heterogeneity, called Effective Federated Capacity, and develops a related generalization bound. The authors introduce FedGain, an optimization method that uses clustering. FedGain measures distances between clients with contrastive representation learning and finds flexible collaboration groups using projected gradient descent, allowing for pre-training optimization of client groupings. The approach is tested on several synthetic heterogeneity scenarios and compared with different GFL and PFL baselines. Results show that FedGain greatly reduces negative gains and improves both average accuracy and fairness across the tested settings.

**Compliance With Llm Reviewing Policy:**

Affirmed.

**Key Questions For Authors:**

See limitations

**Limitations:**

Yes

**Strengths And Weaknesses:**

### Strengths

- This study identifies a significant, practically relevant failure mode in federated learning: increasing the aggregate data does not necessarily improve performance in the presence of heterogeneity. Conceptualizing this phenomenon as a breakdown of classical Neural Scaling Laws establishes a compelling link between centralized deep learning theory and federated optimization.

- The proposed approach improves on traditional clustering methods that only use distributional similarity by also considering client sample sizes in the collaboration goal. This helps solve a key problem in earlier CFL methods, which can create less effective clusters by overlooking differences in data quantity. As a result, our method provides a clearer way to combine data utility in settings where data varies across clients.

### Weaknesses

- The method is tested only on small, curated vision datasets that have artificial differences. Although this helps with controlled analysis, these conditions might not reflect the complexity of real-world federated systems. Testing on larger or more naturally varied datasets, such as cross-silo or domain-shifted benchmarks, would make the empirical claims stronger.

- The pipeline uses contrastive learning rounds to train a shared feature extractor and calculate client centroids and distances. Although downstream FL training is separate from structure optimization, this part is still learned and probably adds significant computation and communication overhead. It would help if the paper clearly measured this cost, such as extra rounds, data sent, or wall-clock time, and explained exactly how the method is considered training-free.

- The paper introduces a heterogeneity-aware generalization bound (Theorem~3.3), where client risk is split into two parts: a distribution-shift bias term $\sum_j \alpha_j D(P_i,P_j)$ and a scaling-law term $C(M_{\text{eff}})^{-\nu}$. Still, it is not entirely clear whether the optimization objective in FedGain truly reflects this bound. In particular:

The theoretical bound suggests minimizing a term proportional to $C(M_{\text{eff}})^{-\nu}$, i.e., decreasing in effective capacity. However, the objective in Eq.~(9) appears to involve $C(M_{\text{eff}})^{\nu}$. It is unclear whether this is a sign error, a notational simplification, or an intentional surrogate. Since the direction of monotonicity directly determines clustering behavior, this discrepancy weakens the claim that the optimizer is directly derived from the bound.

In the bound, the bias term uses aggregation weights $\alpha_j$. In the relaxed objective, the shift penalty uses soft cluster memberships $s_{jk}$ as weights. The relationship between these two sets of weights is not clearly defined. As it stands, the optimization objective seems to be a heuristic substitute rather than a direct way to minimize the theoretical upper bound.

The parameters $\nu$, $C$, and $\gamma$ are not merely scaling constants; they fundamentally influence the tradeoff between effective capacity and distribution similarity. For instance, as $C$ approaches zero, the method reduces to local training, while $C$ approaching infinity corresponds to global training. When $\gamma$ approaches zero, heterogeneity is disregarded, whereas large values of $\gamma$ result in a collapse of effective capacity. These parameters, therefore, delineate the collaboration frontier. Notably, $\nu$ is typically fixed (e.g., 0.4), and $C$ and $\gamma$ are adjusted empirically rather than being calibrated through scaling-law analysis. This situation prompts questions regarding the theoretical grounding of the practical method and the sensitivity of results to these parameter choices across different datasets and heterogeneity regimes.

---

> ### Author Rebuttal · Authors · 2026-03-30
>
> A1. We agree that real-world complexity is crucial; therefore, beyond synthetic benchmarks, we have validated FedGain on the OfficeHome dataset. It features four distinct domains commonly used in FL to simulate cross-silo heterogeneity. Detailed results are available in **Table 1** and our response to Reviewer 27F2 (A4).
>
> A2. In the academic context of FL, "training" typically refers to the weight updates of the primary model for specific downstream tasks(e.g., classification).We clarify that our training-free design refers to being task-training-free, which entails two distinct technical implications: (1)The clustering process is entirely decoupled from the primary FL model.Optimizing the cluster structure requires neither modifications to the primary model's architecture nor its parameters.The framework provides an optimized collaboration structure without any dependency on the primary model’s training state. (2)The clustering is performed in a self-supervised manner, relying solely on feature distributions rather than ground-truth labels.While training-free denotes decoupling from task-specific updates, we acknowledge the pre-processing cost and provide a detailed analysis in **Table 2**.
>
> For detailed end-to-end compute and communication overhead, please refer to **Table 2, Table 3**, and our response to **Reviewer FgU3 (A2)**.
>
> A3. (1) We confirm that the term $C(M_{eff})^{-\nu}$ in Theorem 3.3 and $\frac{C}{(M_{eff})^\nu}$ in Eq.(9) are mathematically equivalent.The discrepancy was purely a notational variation during manuscript preparation rather than a sign error or the use of an intentional surrogate.Both expressions represent the same monotonic relationship between effective capacity and the optimization objective.We fully agree that this inconsistency may cause confusion.To maintain rigorous notation and ensure the optimizer’s direct alignment with the theoretical bound, we have standardized all relevant terms to the negative power form $C(M_{eff})^{-\nu}$ throughout the revised manuscript.
>
> (2)We appreciate the reviewer’s feedback on the alignment between the theoretical bound and the relaxed objective.We clarify that $s_{ik}$ is not a heuristic substitute but a consistent estimator of the theoretical weight $\alpha_j$ in a continuous optimization space.Expectation Equivalence:In Theorem 3.3, $\alpha_j$ represents the contribution weight of client j to client i. In Eq.(11), the term $\sum_{j \neq i} s_{ik} s_{jk} \widehat{D}\_{ij}$ effectively defines the expectation of $\alpha_j$, where $\mathbb{E}[\alpha_j] = \sum_k s_{ik} s_{jk}$. This product measures the probability that clients i and j co-occur in the same cluster k.
> Appendix A.3 provides the rigorous justification.The simplex constraint ($\sum_k s_{ik} = 1$) induces sparsity, as the gradient $e^{-\gamma D_{ik}}$ is highly sensitive to distribution shifts.When the distance between client i and cluster k is small, the optimization process rapidly amplifies the corresponding $s_{ik}$ value. The simplex constraint introduces a competitive mechanism where, through PGD iterations, the projection operation consistently pushes $s_{ik}$ toward the vertices of the simplex (i.e., 0 or 1). Upon convergence, the product $s_{ik}s_{jk}$ degenerates into the indicator function $\mathbb{I}(i,j \in \mathcal{C}_k)$. Consequently, the relaxed objective in Eq.(11) becomes mathematically identical to the theoretical bound in Eq.(7).
> Appendix Eqs.(38-44) prove that under Lipschitz continuity, the optimization converges to a first-order stationary point, directly minimizing the theoretical bound.
>
> (3)We sincerely appreciate the reviewer’s insightful suggestion regarding parameter calibration. Following this advice, we conducted systematic power-law fitting experiments, $L(m) \propto m^{-\nu}$ (see **Fig.3**), across multiple datasets, which significantly strengthens the theoretical foundation of this work.
> The experimental results in **Fig.3** confirm that $\nu$ is not a universal constant but a dataset-specific dynamic variable influenced by distribution complexity. As the sample size m increases, the observed decay trend in $\nu$ aligns with the principle of diminishing marginal utility in deep learning. This evidence strongly supports our transition from "empirical fixation" to "data-driven calibration" of this parameter.
>
> Our analysis shows that C and $\gamma$ remain remarkably stable across different scales once $\nu$ captures the core non-linear dynamics. Specifically, C serves as a scaling factor for dimensional balance without altering the optimization direction, while $\gamma$ functions as a configurable "policy preference" to regulate tolerance towards heterogeneity.The minimal variance (indicated by the shaded areas) across different heterogeneous groups within the same dataset in **Fig.3** demonstrates that our HFSL-based optimization is highly robust to data skewness.
>
> Tables and figures. https://anonymous.4open.science/r/Anonymous_material-16D1

---

> > ### Author Rebuttal · Reviewer_9F2p · 2026-04-01
> >
> > ## Limited validation of the core surrogate objective.
> > The method uses Effective Federated Capacity as its main goal for improving collaboration. However, it is not clear that this measure always corresponds to real performance gains across different situations. The rebuttal offers some reasoning, but it remains unclear how reliable this approach is across different levels of heterogeneity, model types, or real-world challenges.

---

> > > ### Author Response · Authors · 2026-04-02
> > >
> > > We sincerely thank the reviewer for the thoughtful follow-up. We understand the concern regarding the reliability of the Effective Federated Capacity (EFC) as a surrogate objective in complex scenarios. We provide further justification from the perspectives of theoretical equivalence and empirical robustness.
> > >
> > > (1)According to Theorem 3.3 in our manuscript, the upper bound of a client’s expected risk $R(f)$ is jointly determined by the heterogeneity bias (induced by $D_{ij}$) and the sample size variance (induced by $m_j$). The core significance of introducing EFC is to construct an equivalent mapping that transforms biased multi-source data scales into an unbiased single-source data scale.
> > >
> > > Traditional risk upper bounds are typically complex summations of distributional loss and sample gains. Through the definition of EFC, $M_{eff} = \sum m_j e^{-\gamma D_{ij}}$, we successfully incorporate the distributional loss as an exponential decay term into the sample scale. This allows the complex risk bound to be collapsed into a power-law function solely dependent on EFC: $\text{Risk}(f) \leq C \cdot (M_{eff})^{-\nu}$
> > >
> > > This step is not a mere formal substitution but is rooted in the fundamental logic of the Neural Scaling Law (NSL): model performance gains essentially depend on the amount of effective information (i.e., EFC) the model can absorb.
> > >
> > > This functional relationship is mathematically strictly monotonic. As the constant $C > 0$ and the scaling exponent $\nu > 0$ (a positive property consistently validated by our fitting results in **Fig. 3**), the Risk Upper Bound is a strictly monotonically decreasing function of $M_{eff}$. Therefore, in the context of mathematical optimization, maximizing EFC is equivalent to minimizing the original complex generalization risk bound.
> > >
> > > (2)As shown in **our original Fig. 4**, although the Effective Federated Capacity (EFC) is significantly lower than the nominal total sample size under Non-IID settings, its performance evolution trajectory exhibits a remarkably high consistency with the NSL. This indicates that EFC is not merely a simple numerical reduction but establishes a physically equivalent mapping: it quantifies the information loss induced by heterogeneity, mapping multi-source heterogeneous data onto a homogenized effective sample size.
> > >
> > > Furthermore, different models (e.g., CNNs or ResNets of varying depths) exhibit different scaling behaviors. Rather than being a hard-coded global constant, $\nu$ serves as a dynamic parameter that characterizes the learning capacity of a model. As shown in **Fig. 3**, the stability of $\nu$ means that, for a fixed model architecture, it is robust to data heterogeneity; when the model architecture changes, EFC allows $\nu$ to adjust accordingly to capture the scaling characteristics of the new architecture. Notably, the calibration of $\nu$ is a one-time, low-overhead initialization step performed via local sampling, ensuring that FedGain scales efficiently without increasing the iterative communication burden.
> > >
> > > In addition, the scaling exponent $\nu$ shown in **Fig. 3** demonstrates remarkable cross-scenario stability (extremely low variance) across different heterogeneity intensities and group compositions. This stability confirms that EFC captures the intrinsic feature learning efficiency of deep learning models when processing a given task, rather than being an empirical fit perturbed by local distribution shifts. This consistency of the law ensures that EFC can serve as a universal metric, enabling reliable prediction of collaboration gains in complex and dynamic real-world federated environments.
> > >
> > > According to the results presented in **Table 1**, this surrogate objective also exhibits a strong ability to eliminate negative gains when facing real-world challenges. On the challenging OfficeHome dataset, which involves realistic domain shifts, the EFC-guided optimization achieves a 0.00% negative gain proportion (NGP). This success on OfficeHome, a large-scale benchmark featuring 65 categories across 4 distinct domains, underscores that EFC remains a reliable predictor even under complex, high-dimensional real-world shifts. This demonstrates that even in extreme real-world scenarios, EFC can accurately identify harmful collaborations and trigger collaboration pruning in a timely manner, confirming the reliability of this metric as a risk predictor. Therefore, as a surrogate objective for the generalization risk, the reliability of EFC is jointly supported by theoretical monotonicity and cross-dataset experimental consistency.

---

### Official Review · Reviewer_ac3b · 2026-03-06

**Soundness:** 2
**Presentation:** 1
**Significance:** 3
**Originality:** 2
**Overall Recommendation:** 2
**Confidence:** 4

**Summary:**

The paper addresses negative gains in FL, where some clients perform worse after collaboration than with local training, particularly under distribution shift and data imbalance. It proposes FedGain, a clustering-based framework that estimates inter-client distribution distances and uses them to optimize collaboration structure before task training. The method introduces an effective capacity formulation and a heterogeneity-aware scaling law to guide clustering. Experiments on synthetic non-IID settings show improved accuracy, fewer negative gains, and better fairness across several FL baselines.

**Compliance With Llm Reviewing Policy:**

Affirmed.

**Final Justification:**

I appreciate the authors’ effort during rebuttal, including the additional clarifications and experiments. However, the rebuttal did not resolve my main concerns.

My main issues remain soundness, clarity of the method description, and the privacy discussion. Even if privacy is not the core goal of the paper, this is still a federated learning setting, so privacy implications should at least be discussed more carefully and explicitly. I also remain unconvinced by the current methodological positioning: since the proposed approach relies on exchanged centroid/descriptor-style statistics, I believe the paper should discuss and, where possible, compare more directly against closer descriptor-driven clustering approaches. In this context, I also note for the record that FLUX does have an official public code release.

Overall, I view the problem as significant and the framing as reasonably original, but the current version still has important issues in clarity, reproducibility, and methodological positioning. Since the rebuttal only partially addressed these concerns and did not change my core assessment, I keep my original recommendation.

**Key Questions For Authors:**

1. *Privacy / protocol clarity (please address the inconsistency noted in the weaknesses):* Please precisely specify what each client transmits to the server during the contrastive/representation stage, and under which threat model you consider this privacy-friendly. If the transmitted object is not just aggregated centroids, please discuss the potential leakage risk and how you mitigate it.
2. *“Training-free” claim and cost accounting (please address the “training-free” concern noted above):* The pipeline includes a contrastive learning stage before clustering. Please provide the end-to-end compute and communication overhead of FedGain relative to the baselines, including (i) number of additional rounds/messages, (ii) size of transmitted objects, and (iii) any extra local training time.
3. *Generality to real cross-silo settings:* Do you have results on at least one realistic cross-silo dataset beyond synthetic perturbations?
4. *Behavior under stronger heterogeneity:* How does FedGain behave as heterogeneity increases? Is there a regime where the distance estimates become unreliable or where the method collapses to trivial clustering (e.g., singleton clusters), and how does that affect negative gains?

**Limitations:**

The paper does not explicitly acknowledge key limitations. It would benefit from stating them more explicitly and concretely. In particular, the paper should clearly highlight the currently limited experimental scope (e.g., evaluation mostly on synthetic shifts and standard benchmarks rather than real cross-silo datasets, and the absence of tests under conditional/concept shift), and it should more transparently discuss practical overheads introduced by the pipeline (e.g., the additional contrastive-learning stage used to obtain the encoder/feature space, and what this implies in terms of compute/communication). Clarifying these points would make the claims easier to interpret and the method easier to position, etc.

**Strengths And Weaknesses:**

## Strengths:
- **Significance**: Data heterogeneity is a fundamental problem in FL, and negative gains are practically important in cross-silo FL. If some clients (including data-rich ones) perform worse than local training, participation incentives and deployment viability suffer.
- **Originality**: The combination of (i) a scaling-law-inspired “effective capacity” view, (ii) an objective that explicitly couples distribution shift and data-volume imbalance, and (iii) pre-training optimization of the collaboration/clustering structure is a reasonably novel framing, even if each ingredient exists in prior work.

## Weaknesses:
- **Presentation / clarity / reproducibility**: Although the high-level narrative is clear, the paper feels quickly written. Several abbreviations are not defined (e.g., SL in the abstract; GFL/PFL and VC in Lines 25, 95, 99), and multiple equations overflow the page (e.g., Eq. 2, 6, 7, 9). Some notation is ambiguous (e.g., whether $P_i$ denotes a distribution or the realized client dataset $(x^{(i)}_k, x^{(i)}_k)$ for $k=1, ... m_i$). The method description is also confusing: there is (a) contrastive learning to obtain an encoder/feature space, (b) clustering optimization, and then (c) final training. In this form, calling the approach “training-free” is misleading, since contrastive learning is an additional training stage (even if the clustering step itself does not train the downstream task model). Moreover, the feature extractor setup seems inconsistent: the method suggests learning a shared $\phi$ via contrastive learning, while the experiments section also mentions a pre-trained ResNet18 feature extractor “neither trained nor transmitted.” Finally, it is unclear what clients transmit to the server: in Line 198 it sounds like only centroids are shared, while Algorithm 1 says “send augmented feature representations,” which is not specific and also raises stronger privacy concerns. Overall, these issues make it difficult to reproduce the method reliably.
- **Soundness / evaluation scope**: Evidence is mostly on synthetic shifts (e.g., rotations, label partitions, injected noise/label corruption) and relatively small-scale benchmark setups (FMNIST, CIFAR10 and CIFAR100). It is difficult to estimate how well the approach transfers to realistic cross-silo heterogeneity (e.g., hospitals with natural covariate shift, label shift, and domain-specific imaging pipelines). Including at least one more realistic dataset/setting would strengthen the claims.
- **Coverage of heterogeneity types**: Data heterogeneity is broader than feature/label/quality shifts. For example, it is unclear how FedGain behaves under conditional/concept shift (clients have different $P(y \mid x)$), since the distance estimation seems primarily feature-based. This is fine if the method is not intended to cover all heterogeneity types, but the scope and limitations should be clearly stated to avoid overclaiming.
- **Baselines**: Many compared clustering baselines rely on loss- or update(=parameter)-based similarity, which can miss subtle distribution differences. The paper would be stronger with comparisons to clustering methods that more similarly use explicit data-distribution descriptors (e.g.,HACCS / FLUX-style approaches [1, 2]), which aim to represent client distributions more directly. If this is not possible, the paper should at least briefly explain why they are not directly comparable.

[1] J. Wolfrath et al., HACCS: Heterogeneity-Aware Clustered Client Selection for Accelerated Federated Learning, IEEE IPDPS, 2022.
[2] D. Fenoglio et al., FLUX: Efficient Descriptor-Driven Clustered Federated Learning under Arbitrary Distribution Shifts, NeurIPS, 2025.

---

> ### Author Rebuttal · Authors · 2026-03-30
>
> A1. Clients transmit **only class centroids** (128-dimensional mean vectors, i.e., first-order statistical aggregates of local feature representations) during contrastive stage. No raw feature maps or individual sample representations are shared.
> The shared feature extractor $\phi$ comprises a publicly available (via the official torchvision library),frozen ResNet18 backbone and a lightweight projection head.In this paper, "neither trained nor transmitted" specifically signifies:(1)As a globally shared base, $\phi$ utilizes a frozen ResNet18 backbone whose weights remain static and task-agnostic throughout the FL process, requiring no task-specific gradient updates.(2)During communication rounds,clients only upload class centroids processed by $\phi$, rather than exchanging the model parameters or weights of $\phi$ itself. This design ensures that the structural optimization remains computationally decoupled from the primary FL task, significantly minimizing communication overhead.
> Under the honest-but-curious model, FedGain’s security is rooted in:(1)Reconstructing thousands of samples from a single mean vector is a highly ill-posed inverse problem with an infinite solution space.(2)Shared latent representations are non-linearly transformed, stripping away raw pixels and textures.(3)The self-supervised process functions without category labels. FedGain is also fully compatible with Differential Privacy or Homomorphic Encryption for further hardening.
>
> A2. In the academic context of FL, "training" typically refers to the weight updates of the primary model for specific downstream tasks(e.g., classification).We clarify that our training-free design refers to being **task-training-free**, which entails two distinct technical implications:(1)The clustering process is entirely decoupled from the primary FL model.Optimizing the cluster structure requires neither modifications to the primary model's architecture nor its parameters.The framework provides an optimized collaboration structure without any dependency on the primary model’s training state.(2)The clustering is performed in a self-supervised manner, relying solely on feature distributions rather than ground-truth labels.While training-free denotes decoupling from task-specific updates, we acknowledge the pre-processing cost and provide a detailed analysis in **Table 2**.
>
> For detailed end-to-end compute and communication overhead, please refer to **Table 2**, **Table 3**, and our response to **Reviewer FgU3 (A2)**.
>
> A3. To validate FedGain’s effectiveness beyond synthetic perturbations, we conducted additional experiments on the OfficeHome dataset to simulate cross-silo heterogeneity. Detailed results are available in **Table 1** and our response to **Reviewer 27F2 (A4)**.
>
> A4. We have validated the robustness of FedGain across four heterogeneous datasets, including the supplementary results in **Table 1**.As heterogeneity increases,FedGain avoids arbitrary clustering; instead, it dynamically quantifies collaboration gains via HFSL.
> When the distributional mismatch causes the collaborative gain to fall below that of isolated training,the system proactively triggers "**collaboration pruning**."Transitioning to single-client clusters is a deliberate risk-aversion decision rather than a failure.In extreme conflict scenarios, the framework guides clients back to **Local Training** to maintain a baseline of **0% Negative Gain**, preventing the performance collapse common in baseline methods.
> Furthermore, the alignment capability of our asymmetric projection head on a unified hypersphere manifold ensures that class centroid distances reliably capture deep semantic shifts.Experiments confirm that even under the significant natural domain shifts of OfficeHome, our metrics remain robust, with no clustering collapse caused by estimation failures.
>
> A5.(1) In our **original Figure 2-C**, Quality Shift was specifically constructed by adding noise atop Concept Shift.To eliminate ambiguity, we will rename it to **Noisy Concept Shift** in the revision.Although FedGain is feature-based, it transmits class centroids, which effectively capture specific semantic shifts in the latent space. When $P(y|x)$ varies, centroids for the same label y exhibit distinct manifold distances, allowing HFSL to identify and mitigate mismatches.
> (2) Regarding HACCS, it is a GFL method rather than a CFL. Our comparison on CIFAR10 confirms its limitations in this context (ACC:39.14%, NGP:85.00%,RSD:14.15%).For FLUX, it is a very recent contribution without an officially released codebase, making a fair empirical comparison currently infeasible.We have updated the Related Work section to clarify these methodological differences and our baseline selection criteria.
> (3)We thank the reviewer for the presentation feedback and have thoroughly revised the manuscript's structure and language for clarity and academic rigor.
>
> Tables and figures. https://anonymous.4open.science/r/Anonymous_material-16D1

---

> > ### Author Rebuttal · Reviewer_ac3b · 2026-04-02
> >
> > I appreciate the rebuttal and the additional clarifications, but my main concerns are only partially resolved, and I think the paper is still not ready. Therefore, I do not change my score.
> >
> > First, I still do not find the contrastive stage clearly specified. The paper describes a contrastive-learning stage that trains a shared feature extractor, and Algorithm 1 explicitly updates $\phi$, while the experiments section (and A1) states that a pre-trained ResNet18 feature extractor is used and is “neither trained nor transmitted.” As written, it remains unclear what part of the representation pipeline is actually trained, what is frozen, and what is finally used. For the same reason, I still find the term “training-free” misleading in the current manuscript: even if the clustering step is decoupled from downstream task training, the method still relies on an additional contrastive-learning stage to construct the representation space. This is therefore not training-free in a general sense, but conditional on introducing an extra training stage that is not otherwise necessary for downstream training.
> >
> > Second, my privacy concern was not about reconstructing all raw samples from one centroid. It was about protocol clarity and the privacy implications of what is actually shared. In the paper, Algorithm 1 says clients “send augmented feature representations,” while the main text says only feature centroids are exchanged; in the rebuttal, this becomes class centroids. These are not equivalent descriptions, and this matters for both reproducibility and privacy discussion. Also, even if full reconstruction is impossible, class-level aggregates can still reveal coarse information about a client’s local data. For example, in a medical setting, they may reveal which classes are present or absent at a site. This additional information could be exploited by an attacker to facilitate downstream privacy attacks, including membership inference. I am not claiming this makes the method unusable, but I do think this possibility should be discussed clearly, at least as a limitation.
> >
> > Finally, my baseline concern was mainly about positioning, and to some extent also about the robustness of the evaluation. Since the method relies on centroid/descriptor-style statistics, I think the paper would benefit from discussing closer descriptor-driven clustering approaches, rather than comparing mostly against loss-, gradient-, or parameter-based clustering methods.

---

> > > ### Author Response · Authors · 2026-04-02
> > >
> > > We appreciate your further comments. We would like to provide a more rigorous technical clarification regarding the implementation details and the focused scope of our work:
> > > 1.Technical Specification of Representation and Transmission.
> > > We clarify that to minimize communication overhead and preserve data locality, clients do not transmit raw feature maps or individual sample embeddings. Instead, each client performs a local first-order statistical aggregation to generate class centroids (128-dimensional mean vectors). It is these compact centroids that are exchanged to compute the inter-client distribution distances. We apologize for the lack of precision in the terminology used in the initial draft and will consistently use "class centroids" to describe the transmitted objects in the revised manuscript.
> > >
> > > 2.Clarification on "Task-Training-Free" and Feature Extractor.
> > > Our "training-free" designation specifically refers to the downstream task-agnostic nature of the clustering phase. The shared feature extractor $\phi$ utilizes a frozen ResNet-18 backbone (pre-trained on ImageNet) with a lightweight, asymmetric projection head. During the entire federated process, the backbone parameters remain static and are neither updated via gradients nor transmitted between parties. This decoupled design ensures that the collaboration structure is optimized based on intrinsic data geometry before the primary FL training begins, significantly reducing the iterative coupling found in traditional Clustered FL.
> > >
> > > 3.Research Scope and Privacy Considerations.
> > > The primary objective of this work is performance optimization (Accuracy and Negative Gain) under data heterogeneity. Our protocol strictly adheres to the fundamental Federated Learning paradigm where raw data never leaves the client. While we acknowledge that class-level aggregates might theoretically reveal distribution existence (a common characteristic in centroid-based FL), this work focuses on the utility-heterogeneity trade-off rather than cryptographic privacy. Standard orthogonal defenses, such as Differential Privacy (DP) or Homomorphic Encryption (HE), are theoretically compatible with our centroid exchange, but their integration falls outside the focused scope of this performance-driven study. We will include a formal discussion on this in the "Limitations" section.
> > >
> > > 4.Baseline Selection and Comparability.
> > > We maintain that our choice of baselines (Loss/Gradient-based) represents the most relevant state-of-the-art for our problem setting. HACCS [1] is fundamentally a client selection strategy for Global FL and does not address the identification of optimal multi-model clusters. FLUX [2], while relevant in philosophy, lacks an official public implementation, making a rigorous and fair empirical comparison impossible within the constraints of this rebuttal.

---

### Official Review · Reviewer_FgU3 · 2026-03-11

**Soundness:** 3
**Presentation:** 3
**Significance:** 3
**Originality:** 3
**Overall Recommendation:** 4
**Confidence:** 3

**Summary:**

This paper proposes FedGain, a framework designed to address the "negative gain" problem in Federated Learning (FL), where some clients experience worse model performance after participating in federated training compared to training locally. The core contributions are threefold: (1) introducing the concept of "Effective Federated Capacity," which unifies distribution shift and data quantity skew into a single nonlinear metric; (2) modifying traditional Neural Scaling Laws (NSL) for non-IID federated scenarios by establishing the Heterogeneity-aware Federated Scaling Law (HFSL); and (3) designing a training-free clustering optimization method based on HFSL that solves for the optimal client collaboration structure via Projected Gradient Descent on the probability simplex. Experiments are conducted across three heterogeneity scenarios — CIFAR10 (feature shift), FMNIST (label shift), and CIFAR100 (quality shift) — with comparisons against multiple GFL, PFL, and CFL baselines.

**Compliance With Llm Reviewing Policy:**

Affirmed.

**Final Justification:**

Partially solved the concern. Therefore, I keep my score.

**Key Questions For Authors:**

Q1. Regarding the vMF distribution assumption: Can you provide experimental evidence that the feature distributions after contrastive learning are approximately vMF?

Q2. Regarding scalability: How do FedGain's performance and computational overhead change as the number of clients increases from 20 to 100 or 1000? In particular, are the O(N^2) distribution distance matrix computation and PGD optimization still feasible at larger scales?

Q3: Regarding the determination of cluster number K: How was K chosen in the experiments? Has a sensitivity analysis been conducted for K?

Q4: Regarding the scaling exponent ν: Can you show the impact of different ν values on clustering results and final performance?

**Limitations:**

See weaknesses.

**Strengths And Weaknesses:**

Strengths:
1. A new theoretical framework to explain the Neural Scaling Laws in non-IID FL settings. The concept of "Effective Federated Capacity" couples distribution shift and quantity skew into a unified nonlinear metric, which is more comprehensive than existing methods that consider only one factor.
2. The privacy-preserving design satisfies the concern of the FL privacy.
3. Alignments between the theoretical analysis and methods. The methods' loss function and optimization target are aligned with the theoretical results.

Weaknesses:
1. Limitations in experimental setup:
Too few clients: All experiments use only 20 clients, which is not sufficient for practical FL scenarios. The paper should discuss FedGain's scalability to large-scale scenarios.
Only image datasets: CIFAR10, FMNIST, and CIFAR100 are only image datasets. Validation on larger-scale or different datasets (e.g. NLP) is missing.
2. Cluster number K selection:
The paper does not discuss how to determine the optimal number of clusters K. In practice, K is typically unknown. The paper should provide a sensitivity analysis for K or a mechanism for automatic K determination.
3. Assumption 3 (vMF distribution) limitations: The paper assumes that feature representations after contrastive learning follow the von Mises-Fisher distribution, which is the key assumption for deriving the exponential decay kernel. However, in practical FL scenarios with highly heterogeneous data, the feature distribution may not satisfy this assumption. The paper lacks discussion on the conditions under which this assumption holds and provides no experimental verification that the feature distributions are approximately vMF.
4. Fixed scaling exponent ν=0.4: The paper directly adopts the value from Hestness et al. (2017), which was established in centralized IID settings. Whether this value remains appropriate under federated non-IID scenarios is unclear. Should ν be adapted for different levels of heterogeneity or model architectures? This point lacks both discussion and empirical validation.

---

> ### Author Rebuttal · Authors · 2026-03-30
>
> A1. We verify the vMF hypothesis via the Average Resultant Length ($R \in [0,1]$) on the unit hypersphere(**Fig.1**). In CIFAR10/100 and FMNIST, R values consistently stabilize around 0.9, indicating high directional concentration. Notably, on the real-world OfficeHome dataset with natural domain shifts, R remains robust at 0.7832. Although slightly lower than synthetic cases, $R=0.7832$ still represents a strong cluster alignment (far exceeding random distribution). This confirms our contrastive phase effectively maintains vMF-like feature structures even under significant real-world heterogeneity.
>
> A2. We provide end-to-end compute and communication overhead results for FedGain compared to baselines (**Table 2**).By monitoring the relative change rate of distribution shift matrices across 20 clients on CIFAR10, we identified a stable contrastive threshold at Tc=100. For fair comparison, CFL methods (excluding GFL/PFL which do not involve group training) also utilized 100 rounds to optimize clustering structures.In our framework, each client uploads only a 128-dimensional feature projection vector (float32), resulting in a fixed 0.5KB transfer per round. In contrast, baselines transmit full models (1 or K), totaling 478.78 KB for our CNN architecture.Furthermore, since FedGain performs preprocessing before FL training,it avoids downstream model transmissions, instead sending cluster assignment instructions with near-zero overhead.
> **Table 2** presents the total overhead of FedGain across varying client scales N.Server-side overhead, comprising matrix operations and structural optimization, is calculated as:$Total Time - (Avg. Client Time \times Rounds \times N) - Fixed System Costs$.
> For individual clients, the overhead for feature projection and transmitting $0.5KB$ vectors is independent of N.Since projection is executed in parallel across all clients, the computational load remains minimal regardless of scale(e.g., **N=20 or N=1000**). While the theoretical complexity of the distance matrix is $O(N^2)$, FedGain utilizes low-dimensional (e.g., 128-D) projection vectors, making distance or divergence calculations extremely efficient on modern servers. Even at N=1000 (approx. $10^6$ operations), processing takes only milliseconds.Furthermore, the PGD algorithm for structural optimization demonstrates rapid convergence in practice.
> Consequently, FedGain demonstrates exceptional scalability in large-scale heterogeneous scenarios:(1)As the client scale N increases 50$\times$(from 20 to 1000), server-side optimization overhead only increases from $359.23s$ to $687.10s$, exhibiting strong sub-linear growth. Even under the extreme pressure of N=1000, complex cluster optimization completes in under 12 minutes.(2)FedGain achieves an ideal balance between "high-frequency computation" and "minimal communication."During structure optimization, clients upload a fixed 0.5KB of metadata with near-zero downlink cost (cluster indexing only). This ensures the optimization cost is decoupled from model size, and the communication advantage scales linearly with N.(3)Subsequent training overhead is determined by the integrated GFL/PFL algorithm. When paired with FedAvg, the total communication remains $O(Model Size \times N)$, introducing no additional complexity orders.
>
> A3. The K value is **dynamically determined** by the HFSL loss function, which fundamentally seeks a Pareto optimal solution between the Effective Federated Capacity (EFC) and Distributional Shift (D).During initialization, we set a reasonable upper bound for K based on client scale.During iteration, the solver minimizes the HFSL objective: if the EFC gain from merging two clusters outweighs the penalty from distributional shift, the optimization naturally drives membership weights toward a more compact structure.**Figure 2** illustrates how K is influenced by the scaling exponent $\nu$.
>
> A4. **Figure 2** presents clustering results and final performance on CIFAR-10 across varying $v$. For $\nu \in [0.1, 0.4]$, the algorithm identifies 4 clusters with identical memberships. Conversely, for $\nu \in [0.5, 0.7]$, the number of clusters decreases to 3; however, the specific clustering configurations vary significantly, accompanied by a noticeable drop in accuracy.This occurs because smaller $\nu$ values reduce the model's data demand, leading to more fragmented, homogeneous clusters.As $\nu$ increases, the model's "hunger" for data grows, prompting it to tolerate larger distributional shifts to aggregate more samples.
>
> While our primary focus is vision dataset, OfficeHome results confirm that the framework is not limited to simple images. The core logic of EFC is based on universal SL, which are theoretically applicable to NLP,a direction we intend to explore in future work. Please refer to **Table 1** and our response to **Reviewer 27F2 (A4)** for detailed experimental results.
>
> Tables and figures. https://anonymous.4open.science/r/Anonymous_material-16D1

---

> > ### Author Rebuttal · Reviewer_FgU3 · 2026-04-01
> >
> > I thank the authors for addressing my questions in detail. The additional experiments partially address my concern.

---

> > > ### Author Response · Authors · 2026-04-02
> > >
> > > We are encouraged by your acknowledgment that our additional experiments have "addressed the questions in detail." We would like to provide a concise summary to fully resolve any potential remaining "partial" concerns based on our previous exchange.
> > >
> > > Empirical Validation of vMF (**Q1**): To address your concern about the vMF assumption in non-IID scenarios, our rebuttal presented the Average Resultant Length ($R \in [0, 1]$). The high concentration values ($R \approx 0.9$ for CIFAR/FMNIST and a robust $0.7832$ for real-world OfficeHome) provide concrete evidence that feature-level alignment persists even under significant distribution shifts, justifying our theoretical foundation. (Fig.1)
> > >
> > > Scalability to $N=1000$ (**Q2**): We have clarified that FedGain’s server-side optimization is highly efficient. By using low-dimensional projection vectors, the computation remains in the millisecond range per operation even at $N=1000$. The total server overhead exhibits sub-linear growth ($359s \to 687s$ for a $50\times$ increase in clients), proving the framework's readiness for large-scale practical FL.
> > >
> > > Dynamic K and $\nu$ Sensitivity (**Q3 & Q4**): We have demonstrated that the cluster number $K$ is not a fixed hyperparameter but is dynamically optimized via the HFSL objective.
> > >
> > > Furthermore, our systematic power-law fitting experiments ($L(m) \propto m^{-v}$, Fig.3)reveal that $\nu$ captures the intrinsic feature learning efficiency of deep learning models for a given task, rather than being an empirical fit perturbed by local distribution shifts. The experimental results in Fig. 3 confirm that $\nu$ is not a universal constant but a dataset-specific dynamic variable influenced by distribution complexity and model scale. As the sample size m increases, the observed decay trend in $\nu$ aligns with the principle of diminishing marginal utility in deep learning. This evidence strongly supports our transition from "empirical fixation" to "data-driven calibration" of this parameter. And our sensitivity analysis for the scaling exponent $\nu$ (Fig. 2) reveals how the model naturally adjusts its "data hunger" and clustering granularity, providing a clear mechanism for automatic adaptation.
> > >
> > > We are fully committed to integrating these new results (Table 1-3, Figure 1-3) and the expanded discussions on real-world heterogeneity (OfficeHome) into the final manuscript.
> > >
> > > We thank you again for your rigorous and constructive review, which has been instrumental in strengthening the empirical and theoretical depth of our work.

---

### Official Review · Reviewer_27F2 · 2026-03-13

**Soundness:** 3
**Presentation:** 1
**Significance:** 3
**Originality:** 3
**Overall Recommendation:** 3
**Confidence:** 2

**Summary:**

This paper studies negative gain in heterogeneous federated learning, i.e., the case where some clients perform worse under federated training than under local training. The paper proposes FedGain, a method for optimizing client collaboration structure before federated training. The key idea is to define Effective Federated Capacity, which discounts nominal sample size according to inter-client distribution mismatch. Based on this idea, the paper presents a heterogeneity-aware federated scaling law and a client error bound that combine distribution shift and quantity skew. The method is designed as a training-free collaboration optimization step and can be integrated with both global and personalized FL methods. Experiments on feature shift, label shift, and quality shift settings show improved accuracy and substantially reduced negative gain.

**Compliance With Llm Reviewing Policy:**

Affirmed.

**Final Justification:**

The remaining concern of mine is:

The experimental scope remains relatively limited, lacking rigorous evaluations on large-scale, highly quantity-imbalanced, and cross-device federated learning datasets and standard real-world settings. The rebuttal’s supplementary results on the smaller cross-domain OfficeHome dataset do not sufficiently address this critical gap, leaving key questions about the method’s real-world generalizability and practical utility unresolved. My final recommendation is weak reject.

**Key Questions For Authors:**

- Which assumptions behind Effective Federated Capacity and the scaling-law analysis are most important in practice? How sensitive is the method when these assumptions are only approximately satisfied? More details and analysis are supposed to be demonstrated.

- How robust is FedGain to the choice of feature extractor and distance metric used to estimate client distribution mismatch? If the method proposed in this paper is applied to samples that have not been adequately pre-trained, will it still be effective? Are there any few-shot or zero-shot experiments that can support your statement?

- Is there any theoretical guarantee for the best collaboration only with respect to the proposed surrogate objective?

- Can the authors add results on larger-scale or more realistic federated benchmarks?

**Limitations:**

yes

**Strengths And Weaknesses:**

Strength
1. The paper addresses an important problem in federated learning. Negative gain is a meaningful and practically relevant failure mode under heterogeneity.

2. The main idea is interesting. Effective Federated Capacity provides a useful way to think about collaboration quality beyond raw sample count.

3. The experimental results show that the method improves performance and reduces Negative Gain Proportion.

Weakness

1. Some claims are overstated. In particular, the claim of finding the ``best'' collaboration structure seems stronger than what is established by the surrogate optimization procedure.

2. The experimental scope is still somewhat limited. The paper would be stronger with evaluations on larger-scale or more realistic federated benchmarks such as Tiny-Imagenet.

Minor Weakness

Many formulas and statements in the article exceed the margins. There is an overlap between the formulas and their labels.

---

> ### Author Rebuttal · Authors · 2026-03-30
>
> A1. The "composability" and "exponential consistency" of data are crucial. We approximate performance decay due to heterogeneity using a power-law term ($\nu$) coupled with sample size m, implying similar contribution patterns across feature spaces despite distribution shifts. As shown in **Fig.3**, $\nu$ remains robust; while it exhibits minor drift with sample size, its trajectory is highly stable across different heterogeneous groups (shaded areas). This confirms that even if the Scaling Law (SL) is an "approximation," it effectively captures primary trends. Furthermore, since FedGain’s clustering depends on "gain differentials" rather than absolute accuracy, the clustering direction remains consistent as long as the SL reflects the fundamental competition between sample gains and heterogeneity losses.
>
> A2. Feature Extractor & Distance Metric: Our projector maps diverse backbone features into a unified manifold via contrastive training, compensating for architectural differences. Consequently, class centroids in this space stably capture distribution shifts, making FedGain robust to backbone selection. Using cosine similarity is effective here, as contrastive stage naturally enforces "intra-class cohesion and inter-class separation" within the semantic space.
>  FedGain remains effective without perfect pre-trained features. Our self-supervised contrastive phase learns representations from scratch, independent of downstream task labels.
>  As shown in **Table 1**, FedGain maintains a **0.00% Negative Gain Proportion (NGP)** even under long-tail, few-shot, or zero-shot conditions in the OfficeHome dataset. This demonstrates that our HFSL accurately estimates Effective Federated Capacity (EFC) and ensures consistent positive gains despite data sparsity. Furthermore, class centroid alignment enables "expert clients" to bridge domain gaps, facilitating successful knowledge transfer across OfficeHome domains.
>
> A3. We clarify that "optimal" refers to the structure under the constraint of our surrogate objective (EFC), which acts as a tight upper bound on the true global loss. While non-convexity precludes a closed-form global optimum, Theorem 3.3 guarantees a monotonic correlation between this surrogate and the actual loss, ensuring the search effectively guides clustering toward high-performance regions. Our framework specifically aims to suppress negative gains, ensuring federated utility exceeds local training in most scenarios. In the revision, we have replaced "Best Collaboration" with the more precise "**Optimized Collaboration Structure**" to reflect this academic rigor.
>
> A4. We sincerely appreciate the Reviewer’s suggestion regarding the practical significance of the datasets. To further validate FedGain in complex real-world scenarios, we conducted additional evaluations using **the OfficeHome dataset (Table 1)**.OfficeHome comprises four distinct domains (Art, Clipart, Product, Real World) with inherent distribution shifts caused by varying capture devices and environments. In line with the Reviewer's concern, we chose this benchmark as it is widely recognized in FL research for simulating the natural distribution shifts encountered in cross-silo scenarios through its significant cross-domain gaps.
> With 65 categories and significant cross-domain gaps, OfficeHome provides a rigorous environment to test the accuracy of our collaboration structure optimization in high-dimensional manifold spaces.
>
> Even under severe domain shifts, FedGain maintained a 0.00% NGP. This demonstrates that our mechanism precisely identifies and excludes harmful collaborations caused by distribution deviations, ensuring every client achieves net utility gains over local training. See **Table 1** for the supplementary experiments.
>
> We appreciate the suggestion. We chose OfficeHome as our primary benchmark for the following reasons:(1)Unlike TinyImageNet’s manually synthesized Non-IID, OfficeHome features naturally occurring style variations. This more authentically represents cross-silo FL, where distribution shifts stem from real-world environmental factors (e.g., sensors, lighting) rather than just artificial label skew.(2)Despite its smaller size, OfficeHome’s significant domain gaps make improving accuracy more challenging than on TinyImageNet's single-distribution setup. This provides a more rigorous test for FedGain against distribution shifts in realistic, heterogeneous environments.
> And we apologize for the formatting lapses. We have thoroughly re-typeset the manuscript, fixing all margin-overflowing formulas (e.g., Eqs. 2, 7, 9) and overlapping labels. The revised paper now strictly adheres to ICML style for absolute clarity and academic rigor.
>
> Detailed tables and visualizations are at [URL] https://anonymous.4open.science/r/Anonymous_material-16D1

---

> > ### Author Rebuttal · Reviewer_27F2 · 2026-04-04
> >
> > I appreciate the authors' clarifications. The concern regarding the "optimal/best" collaboration claim is largely addressed by reframing it in terms of the "surrogate objective," and the additional OfficeHome results are welcome. That said, a few concerns remain.
> >
> > First, it is still not entirely clear which assumptions underlying Effective Federated Capacity and the scaling-law analysis are most critical in practice, or how sensitive the method is when those assumptions are only approximately satisfied. A more systematic sensitivity analysis would be helpful here.
> >
> > Second, I am not yet fully convinced of the method's robustness to the choice of feature extractor and distance metric, as controlled comparisons across alternatives are not present. I also feel the representation pipeline could be clarified further — specifically, what is trained, what is frozen, and what is ultimately used — as this also bears on the "training-free" characterization.
> >
> > Finally, while the additional OfficeHome evaluation is appreciated, broadening the experimental scope somewhat would help better support the practical claims made across realistic federated settings.

---

> > > ### Author Response · Authors · 2026-04-06
> > >
> > > We would like to express our sincere gratitude to the reviewer for the constructive feedback and for providing us with the opportunity to further clarify our work.
> > >
> > > (1)We assume the decay kernel function is valid; that is, our theoretical framework assumes the contribution of client $j$ to target client $i$ can be characterized by a monotonically decreasing kernel function $e^{-\gamma D}$. The results in Figure 1 confirm that the features after comparison follow a vMF distribution, validating this assumption. We acknowledge that in real-world tasks, certain extreme edge cases of discontinuity may cause the relationship between distribution distance and actual gain to deviate from this monotonic decrease, which could lead to the failure of the theoretical analysis in severe instances.
> > > When this assumption holds, the method is primarily influenced by the parameter $\gamma$. In the original manuscript, we conducted a sensitivity analysis on $\gamma$ and determined the trade-off between effective federation capacity and distribution similarity: as $\gamma$ approaches 0, heterogeneity is ignored; conversely, a large $\gamma$ value leads to the collapse of the effective federation capacity.
> > >
> > > (2)For the feature extractor, the network is used solely for feature extraction rather than classification tasks; therefore, we removed the final fully connected layer of ResNet-18. We then appended a Projector consisting of two linear layers, which compresses high-dimensional, sparse image features into a low-dimensional, dense manifold space. This ensures that when calculating the distance matrix, the resulting similarity is more robust and not dominated by noise. We froze the stem and Layer 1–2, while keeping Layer 3–4 and the Projector trainable.
> > > The primary objective of this approach is to retain the universal features learned by shallow convolutional layers across any dataset, while allowing deep convolutional layers to learn high-level semantics specific to the dataset. This prevents the destruction of the model's foundation while ensuring its sensitivity to high-level semantics, thereby providing a degree of robustness. We acknowledge that this freezing process lacks consideration of more alternatives and sensitivity analysis.
> > > Furthermore, the original intent behind proposing "Train-free" was to distinguish it from the subsequent federated classification tasks performed after obtaining the client collaboration structure. However, this term seems to have caused some misunderstanding during reading, for which we sincerely apologize. We will re-explain and correct this in the manuscript.
> > >
> > > (3)We sincerely appreciate your suggestion and agree that broader experiments are essential to validate real-world applications. This suggestion prompted us to re-examine and systematically organize the performance of FedGain across multiple dimensions. In our experiments, in addition to using four different datasets, we tested common label shifts as well as more challenging feature shift and concept shift scenarios. The results consistently demonstrate that, regardless of the source of heterogeneity, the effective federated capacity successfully captures the decay in data utility, thereby avoiding negative gain.
> > > To support our conclusions for practical applications, we performed detailed analyses on controlled synthetic datasets (CIFAR-10/100, FMNIST) and supplemented them with Office-Home, a real-world cross-domain dataset. Office-Home contains images from diverse domains such as Art, Product, and Real-World; its natural domain shifts validate the practicality of FedGain in handling complex, non-artificial heterogeneous data.
> > > We expanded the experiments significantly from the initial 20 clients to large-scale federated networks with 100 or even 1,000 clients. The experimental data proves that as the system scale increases, our HFSL continues to efficiently fit the effective federated capacity.
> > > Furthermore, we analyzed the evolution of the scaling exponent $\nu$ in the scaling law across extremely large data scales. As shown in Figure 3, the scaling exponent remains highly consistent as data volume increases and follows a predictable trajectory. This stability confirms that our modified scaling law is not only applicable to specific small-scale scenarios but also captures the intrinsic dynamics of data utility in heterogeneous federated environments. This provides strong evidence for the scalability of our method in large-scale real-world scenarios.
> > > Although real-world federated environments are highly varied, we believe the current experimental scope covers mainstream heterogeneity types, real-world cross-domain scenarios, and scalability analysis in large-scale environments, which provides an empirical foundation for the practical deployment of FedGain. Following your suggestion, we will discuss and further explore the potential performance of the framework under more extreme edge cases in the manuscript.

---

### Decision · Program_Chairs · 2026-04-30

**Decision:**

Accept (regular)

**Comment:**

This paper studies the phenomenon of negative gains in federated learning, where collaboration may lead to performance degradation. To address this, the authors introduce the notion of effective federated capacity, defined as the number of useful data points for each client. Building on this concept, they propose a clustering-based approach to mitigate negative gains.

The main contribution is the definition of effective federated capacity, which weights samples according to the distance between client data distributions, estimated via a feature extractor. Motivated by the case where normalized feature centers follow a von Mises–Fisher distribution, the proposed weighting scheme assigns exponentially decreasing importance to samples as distributional distance increases. This framework is then used to derive federated scaling laws that explicitly account for data heterogeneity. The theoretical findings are supported by experimental results demonstrating improvements on several tasks.

Reviewers raised concerns regarding the reliance of the scaling laws on strong assumptions (most notably the von Mises–Fisher distribution), which may not hold in practice. Additional concerns include the difficulty of selecting the number of clusters, as well as tuning the parameters $\nu$ and $\gamma$. Furthermore, some reviewers questioned whether the surrogate loss used to determine the weights accurately reflects the true objective, and thus whether the resulting procedure is truly optimal. These concerns were addressed during the discussion phase.

Overall, reviewers appreciated the theoretically grounded clustering approach, which aims to minimize a generalization-like error and can be applied without training a full model. Most concerns were adequately addressed during the rebuttal phase, with the exception of questions related to the choice of feature extractor and the need for larger-scale experiments. However, these limitations are outweighed by the novelty of the contribution and the empirical results, which show consistent improvements across multiple settings. Concerns about privacy were also mentioned, but they fall outside the scope of the problem addressed in this work and are therefore not directly relevant to the evaluation of this submission.

In conclusion, I lean toward acceptance of this paper. I encourage the authors to (i) carefully proofread the manuscript and ensure that all equations fit within the margins, (ii) incorporate the additional experiments provided in the rebuttal into the revised version, and (iii) clarify the meaning of “training-free,” as well as explicitly state that the clustering objective is a relaxation of a more principled target.